# Exploiting Domain-Specific Features to Enhance Domain Generalization

**Manh-Ha Bui**[1]   **Toan Tran**[1]   **Anh Tuan Tran**[1]   **Dinh Phung**[1,2]
[1] VinAI Research, Vietnam   [2] Monash University, Australia
`{v.habm1, v.toantm3, v.anhtt152, v.dinhpq2}@vinai.io` *

## Abstract

Domain Generalization (DG) aims to train a model, from multiple observed source domains, in order to perform well on unseen target domains. To obtain the generalization capability, prior DG approaches have focused on extracting domain-invariant information across sources to generalize on target domains, while useful domain-specific information which strongly correlates with labels in individual domains and the generalization to target domains is usually ignored. In this paper, we propose meta-Domain Specific-Domain Invariant (mDSDI) - a novel theoretically sound framework that extends beyond the invariance view to further capture the usefulness of domain-specific information. Our key insight is to disentangle features in the latent space while jointly learning both domain-invariant and domain-specific features in a unified framework. The domain-specific representation is optimized through the meta-learning framework to adapt from source domains, targeting a robust generalization on unseen domains. We empirically show that mDSDI provides competitive results with state-of-the-art techniques in DG. A further ablation study with our generated dataset, Background-Colored-MNIST, confirms the hypothesis that domain-specific is essential, leading to better results when compared with only using domain-invariant.

## 1   Introduction and Related work

Domain Generalization (DG) has recently become an important research topic in machine learning due to its real-world applicability and its close connection to the way humans generalize to learn in a new domain. In a DG framework, the learner is trained on multiple datasets collected under different environments without any access to any data on the target domain (1). One of the most notable approaches to this problem is to learn the "domain-invariant" features across these training datasets, with the assumption that these invariant representations are also held in unseen target domains (2; 3; 4; 5; 6). While this has been shown to work well in practice, its key drawback is completely ignoring "domain-specific" information that could aid the generalization performance, especially when the number of source domains increases (7).

For instance, consider the problem of classifying dog or fish images from two source domains: sketch and photo. While the sketch contains a conceptual drawing of the animal, the photo includes their taken picture within a background. In this case, sketch domain-invariant is kept across domains, while domain-specific, e.g., a dog in a house or fish in the ocean, will be discarded due to only existing in the photo domain. However, this background information, when present, could lead to an improvement of the classification performance in target domains due to common association between the objects of its background, and when negligent sketches are hard to distinguish. From a theoretical standpoint, there has also been strong recent evidence to indicate the insufficiency of learning domain-invariant representation for successful adaptation in domain adaptation problems (8; 9). For example,

---

*Correspondence to Manh-Ha Bui: `<hb.buimanhha@gmail.com>`.

35th Conference on Neural Information Processing Systems (NeurIPS 2021).

Zhao et al. (8) has pointed out the degradation in target predictive performance if domain-invariant representations are forced while the marginal label distributions on the source and target domains are overly different.

Utilizing domain-specific features in DG has been widely studied in recent works (e.g., (10; 7)). Ding and Fu (10) introduce multiple domain-specific networks for each domain, then use the structured low-rank constraints to align them with domain-invariant. While this encourages the better transfer of knowledge, its main problem is the requirement of too many domain-specific networks. More recently, Chattopadhyay et al. (7) proposed a masking strategy to disentangle domain-invariant and domain-specific to further boost domain-specific learning, but its key drawback is that domain-invariant/domain-specific representations might not be disentangled since the learning and inferring procedures are performed implicitly (i.e., without any theoretical guarantee) through a mask generalization process. That means it lacks a clear motivation as well as theoretical justifications.

Regarding meta-learning related work, a typical approach involving meta-learning in DG is MLDG (11) that is based on gradient update which simulates train/test domain shift within each mini-batch, mainly to learn transferable weight representations from meta-source domains to quickly adapt to the meta-target domain, and so improve generalization ability. However, their task objective adapts for all representation features which include domain-invariant, since low effectiveness because domain-invariant is stable across domains, pushing to adapt those features might affect the stability of those domain-invariant, leading to a lower generalization performance on the target domain.

To handle these domain-invariant shortcomings, in this paper, we propose a novel theoretically sound DG approach that aims to extract label-informative domain-specific and then explicitly disentangles the domain-invariant and domain-specific representations in an efficient way without training multiple networks for domain-specific. Following the meta-learning idea and mitigating previous work's drawbacks, we apply a meta-learning technique specifically to exploit domain-specific quality which should need to be adapted to unseen domains from source domains. Our contributions in this work are summarized as follows:

- We provide a theoretical analysis based on the information bottleneck principle to point out the limitation of only learning invariant and the importance of domain-specific representation by a certainly plausible assumption.

- We then develop a rigorous framework to formulate elements of domain-invariant/domain-specific representations, in which our key insight is to introduce an effective meta-optimization training framework (11) to learn domain-specific representation from multiple training domains. Without accessing any data from unseen target domains, the meta-training procedure provides a suitable mechanism to self-learn domain-specific representation. We term our approach meta-Domain Specific-Domain Invariant (mDSDI) and provide necessary theoretical verifications for it.

- To demonstrate the merit of the proposed mDSDI framework, we extensively evaluate mDSDI on several state-of-the-art DG benchmark datasets, including Colored-MNIST, Rotated-MNIST, VLCS, PACS, Office-Home, Terra Incognita, DomainNet in addition to our newly created Background-Colored-MNIST for the ablation study to examine the behavior of our mDSDI.

## 2 Methodology

### 2.1 Problem setting and Definitions

Let $\mathcal{X} \subset \mathbb{R}^D$ be the sample space and $\mathcal{Y} \subset \mathbb{R}$ the label space. Denote the set of joint probability distributions on $\mathcal{X} \times \mathcal{Y}$ by $\mathcal{P}_{\mathcal{X} \times \mathcal{Y}}$, and the set of probability marginal distributions on $\mathcal{X}$ by $\mathcal{P}_{\mathcal{X}}$. A domain is defined by a joint distribution $P(x, y) \in \mathcal{P}_{\mathcal{X} \times \mathcal{Y}}$, and let $\mathcal{P}$ be a measure on $\mathcal{P}_{\mathcal{X} \times \mathcal{Y}}$, i.e., whose realizations are distributions on $\mathcal{X} \times \mathcal{Y}$.

Denote $N$ source domains by $S^{(i)} = \{(x_j^{(i)}, y_j^{(i)})\}_{j=1}^{n_i}$, $i = 1, \ldots, N$, where $n_i$ is the number of data points in $S^{(i)}$, i.e., $(x_j^{(i)}, y_j^{(i)}) \overset{iid}{\sim} P^{(i)}(x, y)$ where $P^{(i)}(x, y) \sim \mathcal{P}$; and $x_j^{(i)} \sim P_{\mathcal{X}}^{(i)}$, in which $P_{\mathcal{X}}^{(i)} \sim P_{\mathcal{X}}$. In a typical DG framework, a learning model which is only trained on the set of source domains $\{S^{(i)}\}_{i=1}^N$ without any access to the (unlabeled) data points in the target

domain, arrives at a good generalization performance on the test dataset $S^T = \{(x_j^T, y_j^T)\}_{j=1}^{n_T}$, where $(x_j^T, y_j^T) \overset{iid}{\sim} P^T(x, y)$ and $P^T(x, y) \sim \mathcal{P}$.

First, we present the definition of domain-invariant representation in a latent space $\mathcal{Z}$ under covariate shift assumption (i.e., the conditional distribution $P(y|x)$ is unchanged across the source domains):

**Definition 1.** *A feature extraction mapping $Q : \mathcal{X} \to \mathcal{Z}$ is said to be **domain-invariant** if the distribution $P_Q(Q(X))$ is unchanged across the source domains, i.e., $\forall i, j = 1, \ldots, N, \ i \neq j$ we have $P_Q^{(i)}(Q(X)) \equiv P_Q^{(j)}(Q(X))$, where $P_Q^{(i)}(Q(X)) = P_Q(Q(X)|X \sim P_{\mathcal{X}}^{(i)}), \ i = 1, \ldots, N$. In this case, the corresponding latent representation $Z_I = Q(X)$ is then called the domain-invariant representation (see (3) also).*

As mentioned in the example in the introduction part, the definition 1 reveals that the extracted domain-invariant latent $Z_I$ could be the conceptual drawing of the animal which is shared in both sketch and photo domains. However, when existing background information is taken by a picture such as a house or ocean, it is crucial to take these backgrounds into account because the domain-invariant feature extraction $Q$ might ignore them by only existing in the photo domain. Therefore, we next introduce the definition of domain-specific in latent space as follows:

**Definition 2.** *A feature extraction mapping $R : \mathcal{X} \to \mathcal{Z}$ is said to be **domain-specific** if $\forall i, j = 1, \ldots, N, \ i \neq j$ such that $P_R^{(i)}(R(X)) \neq P_R^{(j)}(R(X))$, where $P_R^{(i)}(R(X)) = P_R(R(X)|X \sim P_{\mathcal{X}}^{(i)}), \ i = 1, \ldots, N$. In this case, given $X \sim P_{\mathcal{X}}^{(i)}$ the corresponding latent representation $Z_S^{(i)} = R(X)$ is then called the domain-specific representation w.r.t. the domain $S^{(i)}$.*

Definition 2 states that for any domain $i$ and $j$, the distributions of the domain-specific latent $P_R^{(i)}(R(X))$ and $P_R^{(j)}(R(X))$ must be completely different. For instance, following our mentioned example, given two fish images from $i$ and $j$ that are sketch and photo domain, the mapping $R(X)$ should extract specific information that only belongs to the domain including the shadow of the fish drawing in the sketch and ocean background information in the photo domain.

To this end, this paper aims to show that only learning domain-invariant will limit the prediction performance and generalization ability. Hence, we next provide a formal explanation for the motivation of learning domain-specific, by showing the potential drawback of only learning domain-invariance in terms of predicting class labels.

## 2.2 A theoretical analysis under the Information bottleneck method

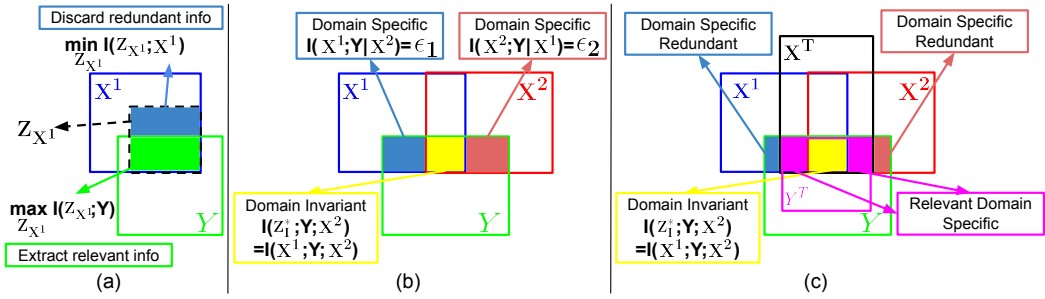

(a)        (b)        (c)

**Figure 1:** Venn diagram showing relationships between source domains represented by $X^1$, $X^2$, target domain represented by $X^T$, and label $Y$. (a) The learning procedure of minimal and sufficient label-related representation in definition 3. (b) Explaining the theorem 1 where the domain-invariant based method provides an inferior prediction performance to our proposed method that incorporates both domain-invariant $I(Z_{I*}; Y)$ and label-related domain-specific values $\epsilon$ made by our assumption 1. (c) A case when the unseen (target) domain has different $X^T$ and $Y^T$, while domain-invariant information is still stable across domains, some domain-specific in source domains become redundant information.

**Notations.** Given three arbitrary random variables $A, B$, and $C$, let us use $I(A; B)$ to represent mutual information between $A$ and $B$; $I(A; B|C)$ to represent conditional mutual information of $A$ and $B$ given $C$; $H(A)$ to represent entropy of $A$; and $H(A|B)$ to represent conditional entropy for random variables $A$ given $B$. For simplicity, we consider the case with two source domains $S^1, S^2$ (the results with multiple source domains can be naturally extended from there). We also define two

corresponding random variables $X^i \sim P_{\mathcal{X}}^{(i)}$, that are sampled from the marginal distribution $P_{\mathcal{X}}$ in the domain $S^i$, $i = 1, 2$.

Figure 1 illustrates all the definitions and assumptions above used for our theoretical verification (in Theorem 1). In particular, in that figure, each of the four colored rectangles represents an individual entropy: $H(X^1)$ for domain $S^1$ is in blue, $H(X^2)$ for domain $S^2$ is in red, $H(X^T)$ for the target domain is in black, and $H(Y)$ for the class label is in green border rectangle.

We first show the ineffectiveness of only learning domain-invariant information when compared with incorporating domain-specific in source domain $S^1$ (and similarly with domain $S^2$). Our justification partly relies on the following assumption about the correlation between the domain-specific representation and the class label:

**Assumption 1.** *(Label-correlated domain-specificity) Assuming that there exists a domain-specific representation $Z_S^{(1)}$ extracted by the deterministic mapping $Z_S^{(1)} = R(X^1)$ in definition 2, which correlates with label in domain $S^1$ such that $I(Z_S^{(1)}; Y | X^2) = I(X^1; Y | X^2) = \varepsilon_1$, where $\varepsilon_1 > 0$ is a constant.*

Assumption 1 indicates that, for the source domain $S^1$, we can learn $Z_S^{(1)} = R(X^1)$ such that $I(Z_S^{(1)}; Y | X^2)$ is strictly positive and equals to $I(X^1; Y | X^2)$, where $I(X^1; Y | X^2)$ is the specific information that correlates with the label in the domain $S^1$, but not in the domain $S^2$ (12). For instance, in the example mentioned in the introduction, if domain $S^1$ is "photo" while $S^2$ is "sketch", the value of $\epsilon_1$ should be positive because the background information such as a house, the ocean also provides information to predict whether the object is a dog or fish without considering its conceptual drawing. This assumption is particularly valid and practically plausible and is demonstrated by several examples observed in our experiments. For instance, for the DomainNet benchmark dataset, in the real-world domain, many bed pictures contain a bed in the room or bike pictures that have bicycles parked on the street. Other examples are in PACS such as dogs in the yard or guitars lying on a table in photo and art domains. These examples are strongly related to assumption 1, in which specific information correlates with labels in a particular domain.

We next present supervised learning frameworks under the umbrella of the information theory (13; 14) and the information bottleneck method (13; 15) that generalizes minimal sufficient statistics to the minimal (i.e., less complexity) and sufficient (i.e, better fidelity) representations. The learning process of such representations is equivalent to solving the following objectives:

**Definition 3.** *(Minimal and sufficient representations with label (14)). Let $Z_{X^1} = G(X^1)$ is the output of a deterministic latent mapping $G$. A representation $Z_{sup}$ is said to be the sufficient label-related representation and $Z_{\sup^*}$ is said to be the minimal and sufficient representation if:*

$$Z_{sup} = \underset{G}{\arg\max} I(Z_{X^1}; Y) \text{ and } Z_{\sup^*} = \underset{Z_{\sup}}{\arg\min} I(Z_{\sup}; X^1) \text{ s.t. } I(Z_{\sup}; Y) \text{ is maximized.}$$

The learning procedure for definition 3 is illustrated in Figure 1: (a). The method is equivalent to employ compressed representations to reduce the complexity (redundant information) of $I(Z_{X^1}; X^1)$ by minimizing and providing sufficient representation to class label $Y$ by maximizing $I(Z_{X^1}; Y)$. Similarly and motivated by multi-view information bottleneck settings (16), we present the objective of learning sufficient (and minimal) representations with domain-invariant information in the below definition:

**Definition 4.** *(Minimal and sufficient representations with domain-invariance (16)). Let $Z_{X^1} = Q(X^1)$ is the output of a deterministic domain invariant mapping $Q$ in the definition 1. Then $Z_I$ is said to be the sufficient domain-invariant representation and $Z_{I^*}$ is said to be the minimal and sufficient representation if:*

$$Z_I = \underset{Q}{\arg\max} I(Z_{X^1}; X^2) \text{ and } Z_{I^*} = \underset{Z_I}{\arg\min} I(Z_I; X^1) \text{ s.t. } I(Z_I; X^2) \text{ is maximized.}$$

Definition 4 introduces a learning strategy for domain-invariance across domains (or views) that preserves shared information across two domains by maximizing $I(Z_{X^1}; X^2)$; and also reduces specificity (redundant information) of the domain $S^1$ by minimizing $I(Z_{X^1}; X^1)$. We next present a lemma about the conditional independence between the latent representation $Z_{X^1}$ and both the label $Y$ and the random variable $X^2$ when $Q$, $R$, and $G$ are deterministic functions of the random variable $X^1$:

**Lemma 1.** *(Determinism (14)) If $P(Z_{X^1}|X^1)$ is a Dirac delta function, then the following conditional independence holds: $Y \perp\!\!\!\perp Z_{X^1}|X^1$ and $X^2 \perp\!\!\!\perp Z_{X^1}|X^1$, inducing a Markov chain $X^2 \leftrightarrow Y \leftrightarrow X^1 \rightarrow Z_{X^1}$.*

The proof of Lemma 1 is provided in Appendix A.1.

Lemma 1 simply states that $Z_{X^1}$ contains no more information than $X^1$. Now, we show the ineffectiveness of only learning domain-invariant approach, based on the existence of the label-related domain-specific in the following theorem:

**Theorem 1.** *(Label-related information with domain-specificity) Assuming that there exists a domain-specific value $\varepsilon_1 > 0$ in domain $S^1$ (see Assumption 1), the label-related representation - based learning approach (i.e., using $Z_{sup}$ and $Z_{sup^*}$) provides better prediction performance than the domain-invariant representation - based method (i.e., using $Z_I$ and $Z_{I^*}$). Formally,*

$$I(X^1; Y) = I(Z_{sup}; Y) = I(Z_{sup^*}; Y) = I(Z_{I^*}; Y) + \varepsilon_1 > I(Z_{I^*}; Y).$$

The proof of Theorem 1 is mainly based on the result of Lemma 1, and is provided in Appendix A.2.

The visualization of Theorem 1 is depicted by Figure 1: (b), where the domain-specific value for domain $S^2$, $\varepsilon_2$ is obtained in the same way as $\varepsilon_1$. It indicates that if an existing domain-specific representation has the positive corresponding information value $\varepsilon$, the domain-invariance-based learning method provides an inferior prediction performance to our proposed method that incorporates both domain-invariant and label-related domain-specific.

Now, Theorem 1 suggests that besides optimizing a domain-invariant mapping $Q$ as usual, we should jointly optimize domain-specific mapping $R$ to achieve a better generalization performance. However, in domain generalization, we are not allowed to access the target domain for training and must use $Q$ and $R$ from source domains. As pointed out in (10; 17; 7), although domain-invariant might be the same because it is unchanged across source domains, there is no guarantee whether this domain-specific information on the source domain is relevant to the target domain while making the prediction. Figure 1: (c) illustrates this case when the target domain has different $X^T$ and $Y^T$, then some extracted domain-specific from source domains become redundant information. Therefore, the next raising question is how to learn domain-invariant and domain-specific effectively.

To handle these shortcomings, we next propose a unified framework that jointly optimizes both $Q$ and $R$ by disentangling their feature representation. In particular, the deterministic mapping $Q$ is optimized by adversarial learning to extract useful domain-invariant features across domains. Meanwhile, by leveraging the transfer weight representations from meta-source domains to adapt to a meta-target domain, we apply meta-learning to deterministic mapping $R$ to force it to extract relevant domain-specific features of the target domain to improve generalization ability.

### 2.3 Algorithm: meta-Domain Specific-Domain Invariant (mDSDI)

So far, we have discussed the main ideas of our proposed method. Here we discuss implementation details for our proposed mDSDI approach.

Figure 2 shows the graphical model and overview of our mDSDI framework. In particular, our unified network consists of the following components: a domain-invariant representation $Z_I = Q_{\theta_Q}(X)$; a domain-specific representation $Z_S = R_{\theta_R}(X)$; a domain discriminator $D_{\theta_{D_I}} : Z_I \rightarrow \overline{1, N}$; a domain classifier $D_{\theta_{D_S}} : Z_S \rightarrow \overline{1, N}$ and a classifier $F_{\theta_F} : Z_I \oplus Z_S \rightarrow \mathcal{Y}$. We also denote domain random variable by $D$, sample space by $\mathcal{D}$, and outcome by $d$.

**Domain-Invariant and Domain-Specific Extraction.** The domain-invariant representation $Z_I$ defined in Definition 1, is obtained by using an adversarial training framework (3), in which the domain discriminator $D_{\theta_{D_I}}$ tries to maximize the prediction probability of the domain label from the latent $Z_I$, while the goal of the encoder $Q_{\theta_Q}$ is to map the sample $X$ to the latent $Z_I$, such that $D_{\theta_{D_I}}$ cannot discriminate the domain of $X$. This task can be performed by solving the following min-max game:

$$\min_{\theta_Q} \max_{\theta_{D_I}} \left\{ L_{Z_I} := -\mathbb{E}_{x,d \sim X,D} \left[ d \log D_I(Q(x)) \right] \right\}. \tag{1}$$

To extract the domain-specific $Z_S$ defined in Definition 2, we propose the use of the domain classifier $D_{\theta_{D_S}}$, that is trained to predict the domain label from $Z_S$. The corresponding parameters $\theta_{D_S}$ and

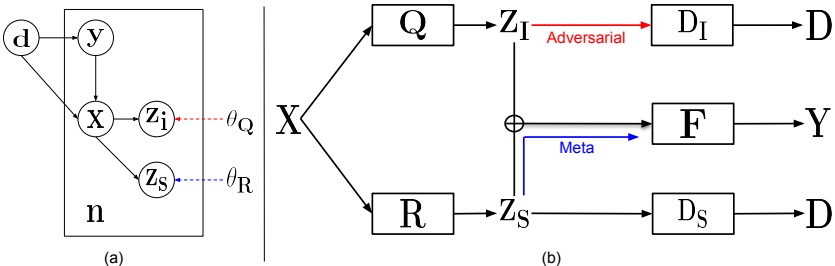

**Figure 2:** The graphical model (a) and overall architecture (b) for our proposed mDSDI, including: domain-invariant $Z_I$ is optimized via adversarial training with domain discriminator $D_I$, domain-specific $Z_S$ is optimized via domain classifier $D_S$, these latent $Z_I$ and $Z_S$ are disentangled by using covariance matrix. To push them to contain label information, these latents are integrated into a classifier $F$ which is optimized via cross-entropy with the label $Y$. To make the model able to adapt specific information from source to unseen domain while still remaining domain-invariance information across domains, we additionally push $Z_S$ through a meta-learning procedure.

$\theta_R$ are, therefore, optimized with the objective function below:

$$\min_{\theta_{D_S},\theta_R} \left\{ L_{Z_S} := -\mathbb{E}_{x,d\sim X,D}\left[d\log D_S(R(x))\right] \right\}. \tag{2}$$

**Disentanglement between Domain-Invariant and Domain-Specific.** The disentanglement condition between two random vectors $Z_I$ and $Z_S$ can be solved by forcing their covariance matrix, denoted by $\mathrm{Cov}(Z_I, Z_S)$ close to 0. A detailed discussion of disentangled two representations is provided in Appendix B.3. The related parameters $(\theta_Q, \theta_R)$ are then updated in the following optimization problem:

$$\min_{\theta_Q,\theta_R} \left\{ L_D := \mathbb{E}_{x\sim X}\left[\|\mathrm{Cov}(Q(x), R(x))\|_2\right] \right\}, \tag{3}$$

where $\|\cdot\|_2$ is the $L_2$ norm.

*Sufficiency of domain-specific and domain-invariant w.r.t. the classification task.* The goal of the classifier $F$ parameterized by $\theta_F$ is to predict the label of the original sample $X$ based on the domain-invariant $Z_I$ and domain-specific $Z_S$, i.e.,

$$\hat{Y} = F_{\theta_F}(Z_I \oplus Z_S), \tag{4}$$

where $\oplus$ denotes the concatenation operation. Then, the training process of $F$ is then performed by solving

$$\min_{\theta_Q,\theta_R,\theta_F} \left\{ L_T := -\mathbb{E}_{x,y\sim X,Y}\left[y\log F(Q(x), R(x))\right] \right\}. \tag{5}$$

**Meta-Training for Domain-Specific Information.** To encourage the domain-specific representation $Z_S$ to adapt information learned from the source domains to the unseen target domain, we introduce the use of meta-learning framework (11), targeting a robust generalization. Note that the domain-invariant feature $Z_I$ remains during the meta-learning procedure. In particular, each source domain $S_m$, $m \in \overline{1,N}$ is split into two sub-domains, namely meta-train $S_{mr}$ and meta-test $S_{me}$. The domain-specific parameters $\theta_R$ and the classifier parameters $\theta_F$ are then jointly optimized as follows:

$$\min_w \left\{ L_{T_m} := f\left(w - \nabla f\left(w, S_{mr}\right), S_{me}\right) \right\}, \tag{6}$$

where $w = (\theta_R, \theta_F)$ and

$$f\left(w, S_m\right) = -\mathbb{E}_{x,y\sim X,Y}\left[y\log F(Z_I, R(x))\right]. \tag{7}$$

**Training and Inference.** The pseudo-code for training and inference processes of our proposed mDSDI framework is presented in Algorithm 1. Each iteration of the training process consists of two steps:

i) First, we integrate the objective functions (1), (2), (3) and (5) to construct an objective function $L_A$ defined as follows:

$$\min_{\theta_Q,\theta_{D_S},\theta_R,\theta_F} \max_{\theta_{D_I}} \left\{ L_A := \lambda_{Z_I} L_{Z_I} + \lambda_{Z_S} L_{Z_S} + \lambda_D L_D + L_T \right\}, \tag{8}$$

where $\lambda_{Z_I}$, $\lambda_{Z_S}$ and $\lambda_D$ are selected as the balanced parameters.

ii) The second step is to employ meta-training to adapt task-related domain-specific from source domains to unseen domains. In each mini-batch, the meta-train and meta-test are split, then the gradient transformation step from meta-train domains to the meta-test domain is performed by solving the optimization problem (6).

---

**Algorithm 1:** Training and Inference processes of mDSDI

---

**Training Input**: Source domain $S^{(i)}$, encoder $Q_{\theta_Q}$, $R_{\theta_R}$, domain classifier $D_{\theta_{D_I}}$, $D_{\theta_{D_S}}$ for $Z_I$, $Z_S$, task classifier $F_{\theta_F}$, batch size $B$, learning rate $\eta$. **Output:** The optimal: $Q_{\theta_Q}^*$, $R_{\theta_R}^*$, $F_{\theta_F}^*$;

**for** $ite = 1 \rightarrow iterations$ **do**

    Sample $S_B$ with a mini-batch $B$ for each domain $S^{(i)}$;

    Compute $L_A$ using Eq. (8) and perform gradient update $\nabla_{\theta_Q, \theta_R, \theta_{D_I}, \theta_{D_S}, \theta_F} L_A$ with $\eta$.;

    **for** $j = 1 \rightarrow N$ *(number of source domains)* **do**

        Split Meta-train $S_{B/j}$, Meta-test $S_j$;

        **Meta-train**: Perform gradient update $\nabla_{\theta_R, \theta_F}$ by minimizing Eq. (7) with $S_{B/j}$ and $\eta$;

        **Meta-test**: Compute $L_{T_m}$ using Eq. (6) with $S_j$ and updated gradient from **Meta-train**;

        **Meta-optimization**: Perform gradient update $\nabla_{\theta_R, \theta_F} L_{T_m}$ with $\eta$;

    **end**

**end**

**Inference Input:** Target domain $S^T$, optimal: $Q_{\theta_Q}^*$, $R_{\theta_R}^*$, $F_{\theta_F}^*$. **Output:** $Y^T$ using Eq. (4);

---

## 3 Experiments

### 3.1 Experimental settings

**Dataset.** To evaluate the effectiveness of the proposed method, we utilize 7 commonly used datasets including: **Colored-MNIST** (18): includes 70000 samples of dimension $(2, 28, 28)$ in binary classification problem with noisy label, from MNIST over 3 domains with noisy rate $d \in \{0.1, 0.3, 0.9\}$, **Rotated-MNIST** (19): contains 70000 samples of dimension $(1, 28, 28)$ and 10 classes, rotated from MNIST over 6 domains $d \in \{0, 15, 30, 45, 60, 75\}$, **VLCS** (20): includes 10729 samples of dimension $(3, 224, 224)$ and 5 classes, over 4 photographic domains $d \in \{$Caltech101, LabelMe, SUN09, VOC2007$\}$, **PACS** (2): contains 9991 images of dimension $(3, 224, 224)$ and 7 classes, over 4 domains $d \in \{$artpaint, cartoon, sketches, photo$\}$, **Office-Home** (21): has 15500 daily images of dimension $(3, 224, 224)$ and 65 categories, over 4 domains $d \in \{$art, clipart, product, real$\}$, **Terra Incognita** (22): includes 24778 wild photographs of dimension $(3, 224, 224)$ and 10 animals, over 4 camera-trap domains $d \in \{$L100, L38, L43, L46$\}$, and **DomainNet** (23): contains 586575 images of dimension $(3, 224, 224)$ and 345 classes, over 6 domains $d \in \{$clipart, infograph, painting, quickdraw, real, sketch$\}$. The detail of each dataset is provided in Appendix C.1.

**Baseline.** Following **DomainBed** (24) settings, we compare our model with 14 related methods in DG which are divided by 5 common techniques, including: ***Standard Empirical Risk Minimization***: Empirical Risk Minimization (**ERM** (25)); ***domain-specific-learning***: Group Distributionally Robust Optimization (**GroupDRO** (26)), Marginal Transfer Learning (**MTL** (1; 27)), Adaptive Risk Minimization (**ARM** (28)); ***Meta-learning***: Meta-Learning for DG (**MLDG** (11)); ***domain-invariant-learning***: Invariant Risk Minimization (**IRM** (18)), Deep CORrelation ALignment (**CORAL** (29)), Maximum Mean Discrepancy (**MMD** (30)), Domain Adversarial Neural Networks (**DANN** (31)), Class-conditional DANN (**CDANN** (32)), Risk Extrapolation (**VREx** (33)); ***Augmenting data***: Inter-domain Mixup (**Mixup** (34; 35; 36)), Style-Agnostic Networks (**SagNets** (37)), Representation Self Challenging (**RSC** (38)). The detail of each method is provided in Appendix C.2.

We use the training-domain validation set technique as proposed in DomainBed (24) for model selection. In particular, for all datasets, we first merge the raw training and validation, then, we run the test three times with three different seeds. For each random seed, we randomly split training and validation and choose the model maximizing the accuracy on the validation set, then compute performance on the given test sets. The mean and standard deviation of classification accuracy

from these three runs are reported. We evaluate generalization performance based on backbones MNIST-ConvNet (24) for MNIST datasets and ResNet-50 (39) for non-MNIST datasets to compare with the mentioned methods. Data-processing techniques, model architectures, hyper-parameters, and changes of objective functions during training are presented in detail in Appendix C.3 C.5. All source code to reproduce results are available at *https://github.com/VinAIResearch/mDSDI*.

## 3.2 Results

**Table 1:** Classification accuracy (%) for all algorithms and datasets summarization. Our mDSDI method achieves highest accuracy on average when comparing 14 popular DG algorithms across 7 benchmark datasets.

| Method | CMNIST | RMNIST | VLCS | PACS | OfficeHome | TerraInc | DomainNet | Average |
|---|---|---|---|---|---|---|---|---|
| ERM (25) | 51.5±0.1 | 98.0±0.0 | 77.5±0.4 | 85.5±0.2 | 66.5±0.3 | 46.1±1.8 | 40.9±0.1 | 66.6 |
| IRM (18) | 52.0±0.1 | 97.7±0.1 | 78.5±0.5 | 83.5±0.8 | 64.3±2.2 | 47.6±0.8 | 33.9±2.8 | 65.4 |
| GroupDRO (26) | 52.1±0.0 | 98.0±0.0 | 76.7±0.6 | 84.4±0.8 | 66.0±0.7 | 43.2±1.1 | 33.3±0.2 | 64.8 |
| Mixup (34; 35; 36) | 52.1±0.2 | 98.0±0.1 | 77.4±0.6 | 84.6±0.6 | 68.1±0.3 | 47.9±0.8 | 39.2±0.1 | 66.7 |
| MLDG (11) | 51.5±0.1 | 97.9±0.0 | 77.2±0.4 | 84.9±1.0 | 66.8±0.6 | 47.7±0.9 | 41.2±0.1 | 66.7 |
| CORAL (29) | 51.5±0.1 | 98.0±0.1 | 78.8±0.6 | 86.2±0.3 | 68.7±0.3 | 47.6±1.0 | 41.5±0.1 | 67.5 |
| MMD (30) | 51.5±0.2 | 97.9±0.0 | 77.5±0.9 | 84.6±0.5 | 66.3±0.1 | 42.2±1.6 | 23.4±9.5 | 63.3 |
| DANN (31) | 51.5±0.3 | 97.8±0.1 | 78.6±0.4 | 83.6±0.4 | 65.9±0.6 | 46.7±0.5 | 38.3±0.1 | 66.1 |
| CDANN (32) | 51.7±0.1 | 97.9±0.1 | 77.5±0.1 | 82.6±0.9 | 65.8±1.3 | 45.8±1.6 | 38.3±0.3 | 65.6 |
| MTL (1; 27) | 51.4±0.1 | 97.9±0.0 | 77.2±0.4 | 84.6±0.5 | 66.4±0.5 | 45.6±1.2 | 40.6±0.1 | 66.2 |
| SagNets (37) | 51.7±0.0 | 98.0±0.0 | 77.8±0.5 | **86.3±0.2** | 68.1±0.1 | **48.6±1.0** | 40.3±0.1 | 67.2 |
| ARM (28) | **56.2±0.2** | **98.2±0.1** | 77.6±0.3 | 85.1±0.4 | 64.8±0.3 | 45.5±0.3 | 35.5±0.2 | 66.1 |
| VREx (33) | 51.8±0.1 | 97.9±0.1 | 78.3±0.2 | 84.9±0.6 | 66.4±0.6 | 46.4±0.6 | 33.6±2.9 | 65.6 |
| RSC (38) | 51.7±0.2 | 97.6±0.1 | 77.1±0.5 | 85.2±0.9 | 65.5±0.9 | 46.6±1.0 | 38.9±0.5 | 66.1 |
| mDSDI (Ours) | 52.2±0.2 | 98.0±0.1 | **79.0±0.3** | 86.2±0.2 | **69.2±0.4** | 48.1±1.4 | **42.8±0.1** | **67.9** |

Table 1 summarizes the results of our experiments on 7 benchmark datasets when compared with mentioned methods. The full result per dataset and domain is provided in Appendix C.4. From these results, we draw three conclusions about our mDSDI model:

**Our mDSDI still preserves domain-invariant information.** We observe in some target domains which have background-less images and assume only contain domain-invariant information such as Colored-MNIST, Rotated-MNIST, or Terra Incognita (similar observation in cartoon or sketch in PACS, clip-art or product in OfficeHome, and quickdraw in DomainNet. Full results in Appendix C.4), our mDSDI model still achieves competitive results with other baselines (e.g., $52.2\%$ in Colored-MNIST, $98.0\%$ in Rotated-MNIST, and $48.1\%$ in Terra Incognita) which are based on domain-invariant-learning techniques such as DANN, C-DANN, CORAL, MMD, IRM, and VREx. Those results demonstrate the effectiveness of our adversarial training technique for extracting domain-invariant features. Furthermore, due to considering disentangled domain-invariant and domain-specific latent, even in situations where the samples do not have domain-specific, our model still performs well by retaining informative domain-invariance.

**Our mDSDI could capture the usefulness of domain-specific information.** In contrast, we observe that in some target domains that have relevant domain-specific with source domains such as landscape background of the object class from photographic pictures in VLCS (similar observation in PACS such as dogs in the yard or guitars lying on a table in photo and art domain, bed in the room or bike parked on the street in the Art and Real-world domain of OfficeHome. Full results in Appendix C.4), mDSDI achieves significantly higher results than other methods (e.g., $79.0\%$ in VLCS, $86.2\%$ in PACS). This means that our domain-invariant features not only support generalization better but also our domain-specific ones cover helpful information in special scenarios such as backgrounds and colors related to objects in the classification task. Moreover, when comparing with other domain-specific based techniques such as GroupDRO, MTL, and ARM, the results showed that domain-specific features learned by meta-training from our model are more helpful than theirs and have captured useful domain-specific features from those object-background relations.

**Extending beyond the invariance view to usefulness domain-specific information is important.** As shown in Table 1, our mDSDI has the highest average number with $67.9\%$ (highlighted with statistically significant according to a $t$-test at a significance level $\alpha = 0.05$). The reason why our model outperforms other baselines could be explained by the fact that their domain-invariant methods are not able to capture domain-specific information, and so have poor performance. Meanwhile, when comparing with other domain-specific based methods, their models only concentrate on domain-specific techniques, and so provide inferior domain-invariant information to our techniques in some

background-less images. In contrast, due to considering disentangled domain-invariant and domain-specific features, and having the right strategy to learn each latent, our model captures both this useful information, hence, outperforms their results. Not only has the highest average number, but our method also dominates other methods on a known large-scale dataset such as $69.2\%$ in Office-Home or $42.8\%$ in DomainNet. This implies that besides the essential combination between domain-invariant and domain-specific, when the number of datasets increases, our method can extract more relevant information for complex tasks, such as classifying 345 classes in DomainNet. These results also mean that our model has a balance between informative domain-invariant and domain-specific features to adapt better to different environments than others, therefore showing the highest average in all settings.

### 3.3 How does mDSDI work?

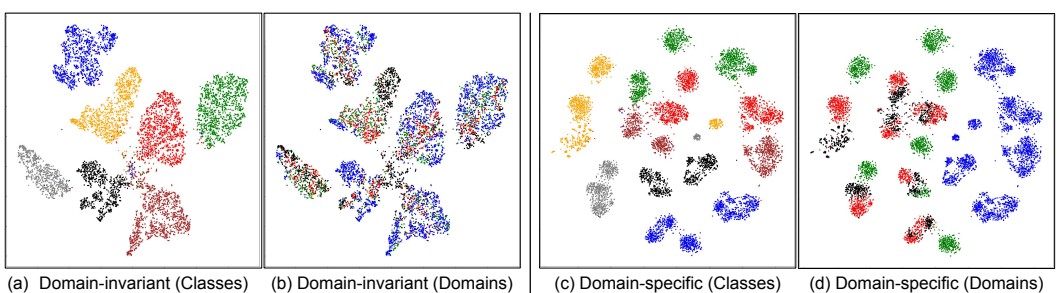

(a) Domain-invariant (Classes)    (b) Domain-invariant (Domains)    (c) Domain-specific (Classes)    (d) Domain-specific (Domains)

**Figure 3:** Feature visualization for domain-invariant: (a): different colors represent different classes; (b): different colors indicate different domains. Feature visualization for domain-specific: (c): different colors represent different classes; (d): different colors indicate different domains. Source domain includes: art (red), cartoon (green), sketch (blue) while target domain is photo (black) in the domain plots. Best viewed in color (Zoom in for details).

To better understand our framework, we visualize the distribution of the learned features with t-SNE (40) to analyze the feature space of domain-invariant and domain-specific. As shown in Figure 3 on PACS Dataset, our domain-invariant extractor can minimize the distance between the distribution of the domains (see Figure 3: (b)). However, these domain-invariant features still make mistakes on the classification task, indicated by a mixture of points from different class labels in the middle (see Figure 3: (a)), and many of these points are from the target domain (black color in the Figure 3: (b)). Meanwhile, the domain-specific representation better distinguishes points by class label (see Figure 3: (c)). More importantly, the photo domain's specific features (black) are close to the art domain (red) (see Figure 3: (d)). This is reasonable because only these two domains include backgrounds related to the object class. It implies that meta-training in our model well learns specific features that can be adapted to the new unseen domain.

### 3.4 Ablation study: Important of mDSDI on the Background-Colored-MNIST dataset

This section examines our system design by checking its performance under different settings in the real scenario with our generated dataset (a similar experiment with PACS benchmark dataset is in Appendix C.6).

**Background-Colored-MNIST dataset.** Figure 4 illustrates our Background-Colored-MNIST, generated from the original MNIST. We assume the domain-invariant is the digit's sketch and design the dataset so that the background color is domain-specific. As a result, on the unseen domain, domain-specific will be useful for the classification task. Specifically, the dataset includes three source domains $d_{tr}$, different by digit's color $\{$red, green, blue$\}$, generated from a subset with 1000 training images for MNIST per each domain. In each source domain, the background color is the same for intra-class images but different across classes. In the target domain, 10000 testing images of MNIST are colored for one target domain $d_{te}$ with digit color $\{$orange$\}$. In this domain, each class's background color is similar to the same class's background color in one of three source domains.

**Importance of mDSDI.** We aim to prove the combination of learning disentangled representation domain-specific, domain-invariant, and meta-training on domain-specific are important in this scenario. To do so, we compare our model under nine settings: learning domain-invariant only (DI),

learning domain-specific only (DS), meta-training on domain-invariant (DI-Meta), meta-training on domain-specific (DS-Meta), a combination of domain-invariant and domain-specific without disentanglement loss $L_D$ (DSDI-Without $L_D$), a combination of domain-invariant and domain-specific without meta-training (DSDI-Without Meta), meta-training on both representation $Z_I$ and $Z_S$ (DSDI-Meta), meta-training on domain-invariant without domain-specific (DSDI-Meta DI) and our proposed framework (mDSDI-Meta DS), which is meta-training on domain-specific without domain-invariant. Table 2 shows that our model is the best setting with $89.7\%$. It proves that combining domain-invariant and domain-specific is crucial by dominating the settings with only domain-invariant or domain-specific (DI, DI-Meta, DS, DS-Meta). Regarding disentangling two representations $Z_I$ and $Z_S$, it is worth noting that without disentanglement loss $L_D$, the model only achieves $81.4\%$, which is lower than mDSDI. It reveals adding disentanglement loss $L_D$ is essential to boost our model performance. Compared with meta-training on both, which only reaches $82.1\%$ and on domain-invariant are $79.0\%$, it implies that meta-training is necessary but only for the domain-specific, and our arguments are reasonable. It also shows that our combination with domain-invariant and domain-specific is not easy: a deep ensemble between two neural networks, but existing a deep-down insight in our framework in DG, when compared with mDSDI-Without Meta which can be seen as a type of ensemble, is only around $80.4\%$.

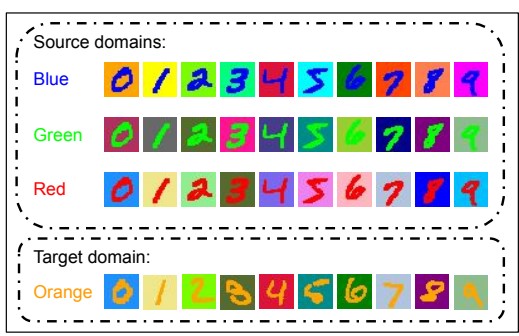

**Figure 4:** Background-Colored-MNIST Dataset, where source domains include $\{$red, green, blue$\}$ digit colors and target domain has $\{$orange$\}$ color.

**Table 2:** Classification accuracy (%) on Background-Colored-MNIST. Ablation study shows impact of domain-invariant when combined with meta-training on domain-specific in our method.

| Method | Accuracy |
|---|---|
| DI | 65.7±4.6 |
| DI-Meta | 63.6±5.1 |
| DS | 70.7±4.8 |
| DS-Meta | 75.3±3.4 |
| DSDI-Without $L_D$ | 81.4±2.6 |
| DSDI-Without Meta | 80.4±1.7 |
| DSDI-Meta | 82.1±1.4 |
| DSDI-Meta DI | 79.0±2.3 |
| mDSDI-Meta DS (Ours) | **89.7±0.8** |

## 4 Conclusion and Discussion

Despite being aware of the importance of domain-specific information, little investigation into the theory and a rigorous algorithm to explore its representation. To the best of our knowledge, our work provides the first theoretical analysis to understand and realize the efficiency of domain-specific information in domain generalization. The domain-specific contains unique characteristics and when combined with domain-invariant information can significantly aid performance on unseen domains. Following our theoretical insights based on the information bottleneck principle, we propose a mDSDI algorithm which disentangles these features. We next introduce the use of the meta-training scheme to support domain-specific to adapt information from source domains to unseen domains. Our experimental results demonstrate mDSDI brings out competitive results with related approaches in domain generalization. In addition, the ablation study with our Background-Colored-MNIST further illustrated and demonstrated the efficiency of combining domain-invariant and domain-specific via our proposed mDSDI. Our theoretical analysis and proposed mDSDI framework can facilitate fundamental progress in understanding the behavior of both domain-invariant and domain-specific representation in domain generalization.

Toward a robustness algorithm that can effectively learn both domain-invariant and domain-specific features, there are certainly many challenges that remain in our paper, for example, a theorem to explain when domain-specific may hurt performance in the unseen domain, a stronger connection between theory in implementation, a method to make two representations to be non-linearly independent as well as a lower computational cost of the covariance matrix. In the future, we plan to continue tackling these challenges to provide a better understanding and learning framework in domain generalization then extending to broader settings of transfer learning. A detailed clarification, discussion, and plausible methods in the future work are provided in Appendix B.

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
