# Exploiting Domain-Specific Features to Enhance Domain Generalization
## (Supplementary Material)

In this supplementary material, we collect proofs and remaining materials that were deferred from the main paper. In Appendix A, we provide the proofs for all the results in the main paper, including: proof of Lemma 1 in Appendix A.1, proof of Theorem 1 in Appendix A.2. In Appendix B, we discuss when domain-specific will effect negatively on the unseen target domain in B.1, clarify the gap between the presented theory and the implemented algorithm in B.2, and continue to discuss the disentanglement technique as well as potential future directions in B.3. In Appendix C, we provide additional information about our experiments, including: sufficient details about the dataset in Appendix C.1, baseline details in Appendix C.2, implementation details to reproduce our experiments in Appendix C.3, further analysis about the results for each benchmark dataset in Appendix C.4, our model's behaviour when training in Appendix C.5, and additional ablation study on the benchmark dataset in Appendix C.6.

## A  Proofs

### A.1  Proof of Lemma 1

*Proof.* When $Z_{X^1}$ is output of deterministic functions from $X^1$, for any A in the sigma-algebra induced by $Z_{X^1}$ we have

$$\mathbb{E}[\mathbb{1}_{Z_{X^1} \in A} | X^1, \{Y, X^2\}] = \mathbb{E}[\mathbb{1}_{Z_{X^1} \in A} | X^1, X^2] = \mathbb{E}[\mathbb{1}_{Z_{X^1} \in A} | X^1],$$

which implies $Y \perp\!\!\!\perp Z_{X^1} | X^1$ and $X^2 \perp\!\!\!\perp Z_{X^1} | X^1$. $\qquad\square$

### A.2  Proof of Theorem 1

*Proof.* The proofs contain two parts. The first one is showing the results for the label-related learned representations and the second one is for the domain-invariant learned representations.

*Part (1). Label-related Learned Representations:* Adapting the Data Processing Inequality (DPI by (41)) in the Markov chain $X^2 \leftrightarrow Y \leftrightarrow X^1 \rightarrow Z_{X^1}$ (Lemma 1), $I(Z_{X^1}; Y)$ is maximized at $I(X^1; Y)$.

Since both label-related learned presentation ($Z_{sup}$ and $Z_{sup^*}$) maximize $I(Z_{X^1}; Y)$, we conclude

$$I(Z_{sup}; Y) = I(Z_{sup^*}; Y) = I(X^1; Y).$$

*Part (2). Domain-invariant Learned Representations:* First, we have

$$I(Z_{X^1}; X^2) = I(Z_{X^1}; Y) - I(Z_{X^1}; Y | X^2) + I(Z_{X^1}; X^2 | Y) = I(Z_{X^1}; Y; X^2) + I(Z_{X^1}; X^2 | Y)$$

and

$$I(X^1; X^2) = I(X^1; Y) - I(X^1; Y | X^2) + I(X^1; X^2 | Y) = I(X^1; Y; X^2) + I(X^1; X^2 | Y).$$

By DPI in the Markov chain $X^2 \leftrightarrow Y \leftrightarrow X^1 \rightarrow Z_{X^1}$ (Lemma 1), we know

- $I(Z_{X^1}; X^2)$ is maximized at $I(X^1; X^2)$
- $I(Z_{X^1}; X^2; Y)$ is maximized at $I(X^1; X^2; Y)$
- $I(Z_{X^1}; X^2 | Y)$ is maximized at $I(X^1; X^2 | Y)$

Since both domain-invariant learned representation ($Z_I$ and $Z_{I^*}$) maximize $I(Z_{X^1}; X^2)$, we have

$$I(Z_I; X^2) = I(Z_{I^*}; X^2) = I(X^1; X^2).$$

Hence,

$$I(Z_I; X^2; Y) = I(Z_{I^*}; X^2; Y) = I(X^1; X^2; Y)$$

and
$$I(Z_I; X^2|Y) = I(Z_{I^*}; X^2|Y) = I(X^1; X^2|Y).$$

Using the result $I(Z_I; X^2; Y) = I(Z_{I^*}; X^2; Y) = I(X^1; X^2; Y)$, we get
$$I(Z_I; Y) = I(X^1; Y) - I(X^1; Y|X^2) + I(Z_I; Y|X^2)$$

and
$$I(Z_{I^*}; Y) = I(X^1; Y) - I(X^1; Y|X^2) + I(Z_{I^*}; Y|X^2).$$

Since $I(Z_{X^1}; X^1) = I(Z_{X^1}; X^1|X^2) + I(Z_{X^1}; X^1; X^2)$, where $I(Z_{X^1}; X^1; X^2) = I(X^1; X^2)$ (Lemma 1) and $I(Z_{X^1}; X^2)$ is maximized at $Z_{I^*}$. Then,
$$I(Z_{X^1}; X^1|X^2) = H(Z_{X^1}|X^2) - H(Z_{X^1}|X^1, X^2),$$

where $H(Z_{X^1}|X^2)$ contains no randomness (no information) as $Z_{X^1}$ being deterministic from $X^1$. Hence, minimizing $I(Z_{X^1}; X^1|X^2)$ equivalents to minimizing $H(Z_{X^1}|X^2)$.

Using the result $I(Z_{I^*}; Y) = I(X^1; Y) - I(X^1; Y|X^2) + I(Z_{I^*}; Y|X^2)$ and due to $H(Z_{X^1}|X^2)$ is minimized at $Z_{I^*}$, $I(Z_{I^*}; Y|X^2) = 0$, we get
$$I(Z_{I^*}; Y) = I(X^1; Y) - I(X^1; Y|X^2)$$

or
$$I(X^1; Y) = I(Z_{I^*}; Y) + I(X^1; Y|X^2).$$

Combining the result in *Part (1)*. $I(Z_{sup}; Y) = I(Z_{sup^*}; Y) = I(X^1; Y)$ and $\epsilon_1 = I(X^1; Y|X^2)$ (Assumption 1), then
$$I(Z_{sup^*}; Y) = I(Z_{I^*}; Y) + \epsilon_1.$$

Moreover, by assuming that $\epsilon_1 > 0$ (Assumption 1), we obtain
$$I(X^1; Y) = I(Z_{sup}; Y) = I(Z_{sup^*}; Y) = I(Z_{I^*}; Y) + \epsilon_1 > I(Z_{I^*}; Y).$$

$\square$

## B    Further discussion

### B.1    Domain-specific features may hurt classification performance in the test domain

Although our theorem 1 suggests that the representation learned by incorporating both domain-invariant and domain-specific has a stronger dependence on labels than that learned by only considering domain-invariant information on source domains $X^1$ and $X^2$. However, there is no guarantee that such representation can generalize well to the "unseen" target domain $X^T$. In the setting of domain generalization, as no target domain data is accessible, the domain-invariant feature may be "invariant" to the source domains only. Moreover, the domain-specific features of the target domain are not able to be extracted accurately if they correlate with other classes than they did in source domains.

In particular, the domain-specific features may hurt performance if they correlate with a different class label in the unseen domain than they did in the source domains (i.e., $I(Z_{I^*}, Y) + \epsilon_1 \neq I(X^T; Y^T)$). Following the example about cow and camel in (18), if in an unseen domain, a camel stands in a field rather than desert, it will hurt our model performance. Another observation is in our ablation study that if the yellow background of number "2" in the source domain becomes the background of number "1" in the target domain, our model's classification accuracy on the class number "1" will drop by 40% in the target domain.

However, the domain-specific features likely will help classification performance in the real world (i.e., $I(Z_{I^*}, Y) + \epsilon_1 = I(X^T; Y^T)$). The correlation between different class labels in the unseen than they did in the source domain is unlikely to appear. For example, camels are more likely to appear in the desert than in a field, and cows are more likely to appear in the field than in the desert. Moreover, our empirical results show several observations in the real world dataset in which domain-specific features of source domains strongly correlate with a class label in both source and target domains. In such case, our mDSDI always have positive results when compared to other domain-invariant based methods, showing our framework is able to select relevant domain-specific information from source domains to generalize well on the target domain.

## B.2 From theory to a unified framework

Despite not exactly minimizing measures of mutual information as discussed in information theory (which causes a high computational cost in DG), our implementation still has a strong connection with the theory section, following these reasons:

- The classifier within standard cross-entropy minimization in Eq. (2) and Eq. (5) has been shown to be similar to minimal and sufficient representation with the label (definition 3) in (14). This is also a fundamental definition of the information bottleneck method, which is proved in (12).

- The domain classifiers within an adversarial training framework in Eq. (1) can be used as a proxy for the minimal and sufficient representations with domain-invariant in definition 4. Because following the definition 4, it maximizes mutual information across source domains and minimizes redundancy specific information in a particular domain. Its optimal solution is similar to the adversarial training framework in Eq. (1).

- In the disentanglement loss in Eq. (3), minimizing Covariance between two random variables is equivalent to minimizing mutual information between them. The reason is: we can derive

$$\min I(Z_I, Z_S) = \min D_{KL}\left(P(Z_I, Z_S), P(Z_I)P(Z_S)\right).$$

  Two random variables X, Y are disentangled (independent) if covariance (X, Y) = 0. Hence, if they are feature vectors $Z_I = [Z_{I^1}, Z_{I^2}, ... Z_{I^m}]$ and $Z_S = [Z_{S^1}, Z_{S^2}, ... Z_{S^n}]$, they will be disentangled (independence) when $\mathrm{Cov}(Z_{I^i}, Z_{S^j}) = 0$ for every $(i, j)$-component in the covariance matrix.

## B.3 Disentangled two representations

The higher the feature dimensionality, the more expensive it is for calculating the Covariance matrix. Specifically, in an experiment in the PACS dataset, if the backbone is Resnet50, we have to store a $2048 \times 2048$ covariance matrix which causes our model to consume around 30GB (GPU).

Given this weakness, we tried an alternative solution with adversarial training to minimize the mutual information between $Z_I$ and $Z_S$. The idea is that we could derive $\min I(Z_I, Z_S) = \min D_{KL}(P(Z_I, Z_S), P(Z_I)P(Z_S))$, and if we shuffled w.r.t. to the index of samples in each mini-batch, we also could disentangle these features without being affected by feature dimension. More specifically, if we create a discriminator $D_{IS}$ to classify representation from $P(Z_I, Z_S)$ as fake samples and representations from $P(Z_I)P(Z_S)$ as real samples. Samples from $P(Z_I, Z_S)$ are obtained by passing the sample $X$ through the encoders $Q$ and $R$ to extract $(Z_I, Z_S)$. Samples from $P(Z_I)P(Z_S)$ are obtained by shuffling the exclusive representations of a batch of samples from $P(Z_I, Z_S)$. The encoder function $Q$ strives to generate exclusive representations $Z_I$ that combined with $Z_S$ look like drawn from $P(Z_I)P(Z_S)$. By minimizing the following objective function:

$$L_{D^{adv}} = \mathbb{E}_{P(Z_I)P(Z_S)}\left[\log D_{IS}(Z_I, Z_S)\right] + \mathbb{E}_{P(Z_I, Z_S)}\left[\log(1 - D_{IS}(Z_I, Z_S))\right].$$

This is equivalent to minimizing the Jensen-Shannon divergence $D_{JS}(P_{Z_I, Z_S} || P(Z_I)P(Z_S))$ and thus the mutual information between $Z_I$ and $Z_S$ is minimized. However, since this is adversarial training, the results had a higher variance in terms of accuracy, so we will try to investigate more in future work.

# C Additional experiments

## C.1 Dataset details

This appendix provides more detail about the dataset in the main paper. Figure 5 visualizes examples for each domain per dataset used in our experiments, including a totally of 7 image datasets widely used for classification tasks in DG and our generated dataset Background-Colored-MNIST in Ablation Study 3.4:

- **Colored-MNIST (18)** includes 70000 samples of dimension $(2, 28, 28)$ in binary classification problem with noisy label, from MNIST over 3 domains with noisy rate $d \in \{0.1,$

| Dataset | Domains | | | | | |
|---------|---------|--|--|--|--|--|
| Colored MNIST | 10% flip | 20% flip | 90% flip | | | |
| |  |  |  | | | |
| | Degree of correlation between color and label | | | | | |
| Rotated MNIST | 0' | 15' | 30' | 45' | 60' | 75' |
| |  |  |  |  |  |  |
| Background-Colored MNIST | Blue | Green | Red | Orange | | |
| |  |  |  |  | | |
| VLCS | Caltech101 | LabelMe | SUN09 | VOC2007 | | |
| |  |  |  |  | | |
| PACS | Art | Cartoon | Photo | Sketch | | |
| |  |  |  |  | | |
| Office Home | Art | Clipart | Product | Photo | | |
| |  |  |  |  | | |
| Terra Incognita | L100 | L38 | L43 | L46 | | |
| |  |  |  |  | | |
| DomainNet | Clipart | Infographic | Painting | QuickDraw | Photo | Sketch |
| |  |  |  |  |  |  |

**Figure 5:** The benchmark dataset summarizations. For each dataset, we pick a single class and show illustrative images from each domain.

$0.2, 0.9\}$. The noisy rate is the correlation ratio between digit and color label. In particular, the construction including assigning a preliminary binary label $\hat{y}$ to the image based on the digit: $\hat{y} = 0$ for digits 0-4 and $\hat{y} = 1$ for 5-9, flipping $\hat{y}$ with probability 0.25 to obtain the final label $y$, flipping $y$ with probability $d$ corresponding to the noisy rate to obtain the color id $z$, and coloring the image red if $z = 1$ or green if $z = 0$. By doing so, if the noise rate in 2 source domains is $d = \{0.1, 0.2\}$ and $d = 0.9$ in the test domain (correlation is reversed in the test environment), then this dataset will allow measuring the invariant learning ability of the model in which the actual invariant is the digit, and the color is just noisy information for the fooling model in source domains.

- **Rotated-MNIST (19)** contains 70000 samples of dimension $(1, 28, 28)$ with 10 classes per each domain, rotated from MNIST over 6 domains $d \in \{0, 15, 30, 45, 60, 75\}$. This data set

does not contain much domain-specific information since the background of binary images is black while only the sketch of the digit is rotated. And so, we assume the domain-invariant is the sketch of the digit.

- **Background-Colored-MNIST (Ours)** contains 1000 training samples of dimension $(3, 28, 28)$ and 10 classes from MNIST per each source domain, colored by digit's color over 3 domains $d_{tr} \in \{\text{red, green, blue}\}$. In addition, the background color is the same for intra-class images but different across classes. There are 10000 testing samples in the target domain, which are colored by $d_{te} \in \{\text{orange}\}$ and each class's background color is similar to the same class's background color in one of three source domains. This data set is used for our ablation study, with the assumption that domain-invariant is the digit's sketch and domain-specific is the background color.

- **VLCS (20)** includes 10729 samples of dimension $(3, 224, 224)$ and 5 classes, over 4 photographic domains $d \in \{\text{Caltech101, LabelMe, SUN09, VOC2007}\}$. The samples from this dataset are collected by taking pictures from real life, hence, contain a lot of domain-specific information on the background such as the landscape of streets where the cars are parked or fields where the bird is eating. However, we observe that in this dataset, besides samples that only contain the object of the class (assume domain-invariant information) without informative backgrounds such as zoom of car or bird, some samples only contain backgrounds (assume domain-specific information) such as houses, ocean or sky landscape.

- **PACS (2)** contains 9991 images of dimension $(3, 224, 224)$ and 7 classes, over 4 domains $d \in \{\text{artpaint, cartoon, sketches, photo}\}$. This is one of the most popular benchmark data sets in DG, while in the artpaint and photo domain contains colored backgrounds that are specific and correlate with the label such as dogs in the yard, the cartoon and sketches only have a white background. And so, we assume the domain-invariant is the object of class and domain-specific is the color and background.

- **Office-Home (21)** has 15588 daily images of dimension $(3, 224, 224)$ and 65 categories, over 4 domains $d \in \{\text{art, clipart, product, real}\}$. Similar to PACS, the majority of images in two domains clipart and product do not include much informative domain-specific information such as color, backgrounds which is related to objects and we assume the domain-invariant is the sketch of the object. Meanwhile, the art and real domains contain more domain-specific features, for instance, the bed usually in a room.

- **Terra Incognita** (22) includes 24788 wild photographs of dimension $(3, 224, 224)$ with 10 animals, over 4 camera-trap domains $d \in \{\text{L100, L38, L43, L46}\}$. This dataset contains photographs of wild animals taken by camera traps; camera trap locations are different across domains. Since these cameras are static, different animals still have the same background, or in other words, there is no correlation between background and animal label in this dataset. Hence, lacks domain-specific features and mainly contains domain-invariant information of the object animal.

- **DomainNet (23)** contains 596006 images of dimension $(3, 224, 224)$ and 345 classes, over 6 domains $d \in \{\text{clipart, infograph, painting, quickdraw, real, sketch}\}$. This is the biggest dataset in terms of the number of samples and classes. The two domains: quickdraw and sketch, only contain conceptual drawings of the object, and so, we assume only having domain-invariant. In contrast, 4 domains: clipart, infographic, painting, and photo have more domain-specific information such as colors, backgrounds, text to describe the object.

## C.2 Baseline details

This appendix provides an exhaustive literature review about 14 related methods in DG which are used to make comparisons with our model, divided by 5 common techniques:

***Standard Empirical Risk Minimization***:

- Empirical Risk Minimization (**ERM** (25)) aggregates all the source domain data together, minimized with cross-entropy for classification loss.

***Domain-specific learning***:

- Group Distributionally Robust Optimization (**GroupDRO** (26)) performs ERM while increasing the importance of domains by weighing mini-batches of the training distribution proportional with larger errors.
- Marginal Transfer Learning (**MTL** (1; 27)) estimates a kernel mean embedding per domain, passed as a second argument to the classifier. Then, these embeddings are estimated using single test examples at test time (only applicable when using RKHS-based learners).
- Adaptive Risk Minimization (**ARM** (28)) an extension of MTL where a separate CNN computes the domain embedding, appended to the input images as additional channels.

*Meta-learning*:

- Meta-Learning for DG (**MLDG** (11)) is the first proposed meta-learning strategy that splits meta train/test and performs gradient update each minibatch, this makes the model trained on one domain to perform well on another domain.

*Domain-invariant learning*:

- Invariant Risk Minimization (**IRM** (18)) learns invariant feature representation such that the optimal linear classifier on top of that representation matches across domains.
- Deep CORrelation ALignment (**CORAL** (29)) matches the mean and covariance (second-order statistics) of features across training domain distributions at some level of representation.
- Maximum Mean Discrepancy (**MMD** (30)) employs the adversarial technique and the maximum mean discrepancy (MMD (42)) criteria to align feature distribution across domain.
- Domain Adversarial Neural Networks (**DANN** (31)) uses an adversarial network to learn feature representation that matches across domains.
- Class-conditional DANN (**CDANN** (32)) is a variant of DANN that matches the feature conditional distributions across domains, for all class labels to enable alignment of multimodal distributions.
- Risk Extrapolation (**VREx** (33)) approximates IRM to reduce the variance of error averages across domains.

*Augmenting data*:

- Inter-domain Mixup (**Mixup** (34; 35; 36)) performs ERM on linear interpolations of examples from random pairs of domains and their labels.
- Style-Agnostic Networks (**SagNets** (37)) promote representations of data that ignore image style and focus on content.
- Representation Self Challenging (**RSC** (38)) learns robust neural networks by iteratively dropping out (challenging) the most activated (important) features.

### C.3 Implementation details

In this appendix, we describe the data-processing techniques, neural network architectures, hyper-parameters, and details for reproducing our experiments. We use similar settings from DomainBed (24) for a fair comparison.

**Data processing techniques.** For experiments related to the MNIST dataset, we receive an image with input size $28 \times 28$ x $d$ pixels (where $d$ is the image dimension, in which $d = 2$ for Colored-MNIST, $d = 1$ for Rotated-MNIST, and $d = 3$ for Background-Colored-MNIST), and divide all the digits evenly among domains. For the remaining datasets, we augment training data using the following protocol: crops of random size and aspect ratio, resizing to $224 \times 224$ x 3 pixels, random horizontal flips, random color jitter, grayscaling the image with $10\%$ probability, and normalization using the ImageNet channel statistics.

**Architectures and hyper-parameters.** We list the details of the backbone network, value of hyper-parameters used for each dataset in Table 3. We optimize all models using Adam (43) optimizer and employ the training-domain validation set technique for model selection. In particular, for all

datasets, we first merge the raw training, validation, and test-set, then, we run the test three times with three different seeds. For each random seed, we randomly split training and validation from each source domain into 80% and 20% splits. and choose the model maximizing the accuracy on the validation set, then compute performance on the domain test-sets after 5000 iterations.

For experiments related to the MNIST dataset, we use MNIST-ConvNet backbone that have the structure following:

*Conv2D (in=d, out=64) → Relu → GroupNorm (groups=8) → Conv2D (in=64, out=128, stride=2) → ReLU → GroupNorm (8 groups) → Conv2D (in=128, out=128) → ReLU → GroupNorm (8 groups) → Conv2D (in=128, out=128) → ReLU → GroupNorm (8 groups) → Global average-pooling.*

For the remaining datasets, we finetune the ResNet-50 model (39) which is pre-trained on ImageNet to avoid the inconsistent choice of network architecture in prior works. We customize the "ResNet-50" by replacing the final (softmax) layer and fine-tune the entire network. Since mini-batches from different domains follow different distributions, batch normalization degrades DG algorithms (44). Therefore, we freeze all batch normalization layers before fine-tuning.

**Table 3:** Condition architectures, hyper-parameters, and their default values in our experiments.

| Condition | Hyper-parameters | Default value |
|---|---|---|
| MNIST-ConvNet | learning rate | 0.001 |
| | batch size | 64 |
| | generator learning rate | 0.001 |
| | discriminator learning rate | 0.001 |
| ResNet | learning rate | 0.00005 |
| | batch size | 32 |
| | batch size (if ARM) | 8 |
| | generator learning rate | 0.00005 |
| | discriminator learning rate | 0.00005 |
| DANN, C-DANN | lambda | 1.0 |
| | discriminator weight decay | 0 |
| | discriminator steps | 1 |
| | discriminator width | 256 |
| | discriminator depth | 3 |
| | discriminator dropout | 0 |
| | discriminator grad penalty | 0 |
| | Adam $\beta_1$ | 0.5 |
| DRO | eta | 0.01 |
| IRM | labmda | 100 |
| | warmup iterations | 500 |
| Mixup | alpha | 0.2 |
| MLDG | beta | 1 |
| MMD | gamma | 1 |
| MTL | ema | 0.99 |
| RSC | feature drop percentage | 1/3 |
| | batch drop percentage | 1/3 |
| SagNets | adversary weight | 0.1 |
| VREx | lambda | 10 |
| | warmup iterations | 500 |
| mDSDI | domain-invariant weight | 1.0 |
| | domain-specific weight | 1.0 |
| | adversary weight | 1.0 |

**Dataset, source code, and computing system.** The source code is provided in the zip file, including scripts to download the dataset, setup for environment configuration, our provided code, and extending

code from DomainBed (24) library (detail in README.md). We run the code on a single GPU: NVIDIA DGX-1 Tesla A100-SXM4-40GB with 12 CPUs: Intel(R) Core(TM) i7-8700 CPU @ 3.20GHz, RAM: 32GB, and require 40GB available disk space for storage.

## C.4 Empirical result details

In this appendix, we show our full results and explain them in more detail when compared with other baseline methods in each benchmark dataset.

**Table 4:** Classification accuracy (%) on Colored-MNIST.

| Method | 10% flip | 20% flip | 90% flip | Average |
|---|---|---|---|---|
| ERM (25) | 71.7±0.1 | 72.9±0.2 | 10.0±0.1 | 51.5 |
| IRM (18) | 72.5±0.1 | 73.3±0.5 | 10.2±0.3 | 52.0 |
| GroupDRO (26) | 73.1±0.3 | 73.2±0.2 | 10.0±0.2 | 52.1 |
| Mixup (34; 35; 36) | 72.7±0.4 | 73.4±0.1 | 10.1±0.1 | 52.1 |
| MLDG (11) | 71.5±0.2 | 73.1±0.2 | 9.8±0.1 | 51.5 |
| CORAL (29) | 71.6±0.3 | 73.1±0.1 | 9.9±0.1 | 51.5 |
| MMD (30) | 71.4±0.3 | 73.1±0.2 | 9.9±0.3 | 51.5 |
| DANN (31) | 71.4±0.9 | 73.1±0.1 | 10.0±0.0 | 51.5 |
| CDANN (32) | 72.0±0.2 | 73.0±0.2 | 10.2±0.1 | 51.7 |
| MTL (1; 27) | 70.9±0.2 | 72.8±0.3 | **10.5±0.1** | 51.4 |
| SagNets (37) | 71.8±0.2 | 73.0±0.2 | 10.3±0.0 | 51.7 |
| ARM (28) | **82.0±0.5** | **76.5±0.3** | 10.2±0.0 | **56.2** |
| VREx (33) | 72.4±0.3 | 72.9±0.4 | 10.2±0.0 | 51.8 |
| RSC (38) | 71.9±0.3 | 73.1±0.2 | 10.0±0.2 | 51.7 |
| mDSDI (Ours) | 73.4±0.2 | 73.1±0.3 | 10.1±0.2 | 52.2 |

**Colored-MNIST.** Table 4 compares our results with the mentioned baseline on the Rotate-MNIST dataset. The average results show our model is not able to achieve a higher result than other methods, especially when compared to ARM (28) with 56.2% on average while our result is only 52.2%. However, it is worth noticing that if the unseen domain has a label-digit correlation which is reversed with source domains (i.e., the unseen domain is 90% flip color), the performance of all models including ARM (28) and IRM (18) (the original paper proposed this dataset), drop significantly, only having around 10% accuracy. This implies that not only our mDSDI, but also all models still concentrate on the color features in this challenging dataset. The color will hurt model performance in this dataset because it will be flipped randomly with a higher probability in the "90% flip color" domain. Therefore, the model should not rely on color features and only focus on digits.

**Table 5:** Classification accuracy (%) on Rotated-MNIST.

| Method | 0 | 15 | 30 | 45 | 60 | 75 | Average |
|---|---|---|---|---|---|---|---|
| ERM (25) | 95.9±0.1 | 98.9±0.0 | 98.8±0.0 | 98.9±0.0 | 98.9±0.0 | 96.4±0.0 | 98.0 |
| IRM (18) | 95.5±0.1 | 98.8±0.2 | 98.7±0.1 | 98.6±0.1 | 98.7±0.0 | 95.9±0.2 | 97.7 |
| GroupDRO (26) | 95.6±0.1 | 98.9±0.1 | 98.9±0.1 | 99.0±0.0 | 98.9±0.0 | **96.5±0.2** | 98.0 |
| Mixup (34; 35; 36) | 95.8±0.3 | 98.9±0.0 | 98.9±0.0 | 98.9±0.0 | 98.8±0.1 | 96.5±0.3 | 98.0 |
| MLDG (11) | 95.8±0.1 | 98.9±0.1 | 99.0±0.0 | 98.9±0.1 | 99.0±0.0 | 95.8±0.3 | 97.9 |
| CORAL (29) | 95.8±0.3 | 98.8±0.0 | 98.9±0.0 | 99.0±0.0 | 98.9±0.1 | 96.4±0.2 | 98.0 |
| MMD (30) | 95.6±0.1 | 98.9±0.1 | 99.0±0.0 | 99.0±0.0 | 98.9±0.0 | 96.0±0.2 | 97.9 |
| DANN (31) | 95.0±0.5 | 98.9±0.1 | 99.0±0.0 | 99.0±0.1 | 98.9±0.0 | 96.3±0.2 | 97.8 |
| CDANN (32) | 95.7±0.2 | 98.8±0.0 | 98.9±0.1 | 98.9±0.1 | 98.9±0.2 | 96.1±0.3 | 97.9 |
| MTL (1; 27) | 95.6±0.1 | 99.0±0.1 | 99.0±0.0 | 98.9±0.1 | 99.0±0.1 | 95.8±0.2 | 97.9 |
| SagNets (37) | 95.9±0.3 | 98.9±0.1 | 99.0±0.1 | **99.1±0.0** | 99.0±0.1 | 96.3±0.1 | 98.0 |
| ARM (28) | **96.7±0.2** | **99.1±0.0** | 99.0±0.0 | 99.0±0.1 | 99.1±0.1 | 96.5±0.4 | **98.2** |
| VREx (33) | 95.9±0.2 | 99.0±0.1 | 98.9±0.1 | 98.9±0.1 | 98.7±0.1 | 96.2±0.2 | 97.9 |
| RSC (38) | 94.8±0.5 | 98.7±0.1 | 98.8±0.1 | 98.8±0.0 | 98.9±0.1 | 95.9±0.2 | 97.9 |
| mDSDI (Ours) | 96.0±0.1 | 98.8±0.1 | **99.1±0.0** | 98.9±0.0 | **99.2±0.1** | 96.2±0.1 | 98.0 |

**Rotated-MNIST.** In Table 5, we compare our results with the mentioned baseline on the Rotated-MNIST dataset. It can be seen that this dataset only contains domain-invariant information while the domain-specific information is limited due to background-less images in the MNIST dataset. However, our mDSDI model still has a competitive result with 98% on average when compared with

other methods that concentrate on learning domain-invariant techniques such as IRM, Corral, MMD, DANN, CDANN, or VREx. This proves that our model still preserves domain-invariant information and the effectiveness of our adversarial training technique for extracting these features.

**Table 6:** Classification accuracy (%) on VLCS.

| Method | C | L | S | V | Average |
|---|---|---|---|---|---|
| ERM (25) | 97.7±0.4 | 64.3±0.9 | 73.4±0.5 | 74.6±1.3 | 77.5 |
| IRM (18) | 98.6±0.1 | 64.9±0.9 | 73.4±0.6 | 77.3±0.9 | 78.5 |
| GroupDRO (26) | 97.3±0.3 | 63.4±0.9 | 69.5±0.8 | 76.7±0.7 | 76.7 |
| Mixup (34; 35; 36) | 98.3±0.6 | 64.8±1.0 | 72.1±0.5 | 74.3±0.8 | 77.4 |
| MLDG (11) | 97.4±0.2 | 65.2±0.7 | 71.0±1.4 | 75.3±1.0 | 77.2 |
| CORAL (29) | 98.3±0.1 | 66.1±1.2 | 73.4±0.3 | 77.5±1.2 | 78.8 |
| MMD (30) | 97.7±0.1 | 64.0±1.1 | 72.8±0.2 | 75.3±3.3 | 77.5 |
| DANN (31) | **99.0±0.3** | 65.1±1.4 | 73.1±0.3 | 77.2±0.6 | 78.6 |
| CDANN (32) | 97.1±0.3 | 65.1±1.2 | 70.7±0.8 | 77.1±1.5 | 77.5 |
| MTL (1; 27) | 97.8±0.4 | 64.3±0.3 | 71.5±0.7 | 75.3±1.7 | 77.2 |
| SagNets (37) | 97.9±0.4 | 64.5±0.5 | 71.4±1.3 | 77.5±0.5 | 77.8 |
| ARM (28) | 98.7±0.2 | 63.6±0.7 | 71.3±1.2 | 76.7±0.6 | 77.6 |
| VREx (33) | 98.4±0.3 | 64.4±1.4 | 74.1±0.4 | 76.2±1.3 | 78.3 |
| RSC (38) | 97.9±0.1 | 62.5±0.7 | 72.3±1.2 | 75.6±0.8 | 77.1 |
| mDSDI (Ours) | 97.6±0.1 | **66.4±0.4** | **74.0±0.6** | **77.8±0.7** | **79.0** |

**VLCS.** In Table 6, we compare our model's performance on VLCS dataset. It shows mDSDI archives the highest score on average with 79.0%, having competitive results on Caltech101 and dominating other baselines on three domains, including LabelMe, Sun09, and VOC2007. Interestingly, we observe that there are many samples in this dataset that miss the object and only contain a background such as an image of the bird but only having a sky picture or car image but only contain houses in the city. Therefore, the reason why our model outperforms other baselines could be explained by the fact that their domain-invariant method could not capture this domain-specific information (sky, houses), and so have a poor performance. Meanwhile, when comparing with other domain-specific based methods, the reason for our higher results would be that their method only concentrates on domain-specific techniques, and so, in some background-less images (e.g., only bird, car), these methods provide inferior domain-invariant information to our techniques. In contrast, due to considering disentangle domain-invariant and domain-specific features, our model captures both this useful information, hence, outperforms their results.

**Table 7:** Classification accuracy (%) on PACS.

| Method | A | C | P | S | Average |
|---|---|---|---|---|---|
| ERM (25) | 84.7±0.4 | **80.8±0.6** | 97.2±0.3 | 79.3±1.0 | 85.5 |
| IRM (18) | 84.8±1.3 | 76.4±1.1 | 96.7±0.6 | 76.1±1.0 | 83.5 |
| GroupDRO (26) | 83.5±0.9 | 79.1±0.6 | 96.7±0.3 | 78.3±2.0 | 84.4 |
| Mixup (34; 35; 36) | 86.1±0.5 | 78.9±0.8 | 97.6±0.1 | 75.8±1.8 | 84.6 |
| MLDG (11) | 85.5±1.4 | 80.1±1.7 | 97.4±0.3 | 76.6±1.1 | 84.9 |
| CORAL (29) | **88.3±0.2** | 80.0±0.5 | 97.5±0.3 | 78.8±1.3 | 86.2 |
| MMD (30) | 86.1±1.4 | 79.4±0.9 | 96.6±0.2 | 76.5±0.5 | 84.6 |
| DANN (31) | 86.4±0.8 | 77.4±0.8 | 97.3±0.4 | 73.5±2.3 | 83.6 |
| CDANN (32) | 84.6±1.8 | 75.5±0.9 | 96.8±0.3 | 73.5±0.6 | 82.6 |
| MTL (1; 27) | 87.5±0.8 | 77.1±0.5 | 96.4±0.8 | 77.3±1.8 | 84.6 |
| SagNets (37) | 87.4±1.0 | 80.7±0.6 | 97.1±0.1 | **80.0±0.4** | **86.3** |
| ARM (28) | 86.8±0.6 | 76.8±0.5 | 97.4±0.3 | 79.3±1.2 | 85.1 |
| VREx (33) | 86.0±1.6 | 79.1±0.6 | 96.9±0.5 | 77.7±1.7 | 84.9 |
| RSC (38) | 85.4±0.8 | 79.7±1.8 | 97.6±0.3 | 78.2±1.2 | 85.2 |
| mDSDI (Ours) | 87.7±0.4 | 80.4±0.7 | **98.1±0.3** | 78.4±1.2 | 86.2 |

**PACS.** Table 7 compares our model with other methods on PACS dataset. It shows that if the target domain is either photo or art, which include more informative backgrounds such as dogs in the yard or guitars lying on a table, mDSDI achieved 98.1% and 87.7% respectively. It is higher than MLDG, which also uses the meta-training technique, but for all representations including domain-invariant. This reveals that although meta-training is essential in DG, it is only for domain-specific features

which need to be adapted to new domains. Moreover, when comparing with other domain-specific based techniques such as GroupDRO, MTL, and ARM, the results showed that domain-specific features from our model are more helpful than theirs.

Meanwhile, due to still considering disentangled domain-invariant features, in two remaining domains of cartoon and sketch, which have a white background, our model still achieves 80.4% and 78.4%. It shows competitive results with other baselines that are based on domain-invariant learning such as DANN, C-DANN, CORAL, MMD, IRM, and VREx. It can be explained by the fact that we only consider meta-training on the domain-specific features, and so still retain informative domain-invariant. As a result, our mDSDI achieved a competitive accuracy with 86.2% on average across test domains on PACS.

**Table 8:** Classification accuracy (%) on Office-Home.

| Method | A | C | P | R | Average |
|---|---|---|---|---|---|
| ERM (25) | 61.3±0.7 | 52.4±0.3 | 75.8±0.1 | 76.6±0.3 | 66.5 |
| IRM (18) | 58.9±2.3 | 52.2±1.6 | 72.1±2.9 | 74.0±2.5 | 64.3 |
| GroupDRO (26) | 60.4±0.7 | 52.7±1.0 | 75.0±0.7 | 76.0±0.7 | 66.0 |
| Mixup (34; 35; 36) | 62.4±0.8 | 54.8±0.6 | **76.9±0.3** | 78.3±0.2 | 68.1 |
| MLDG (11) | 61.5±0.9 | 53.2±0.6 | 75.0±1.2 | 77.5±0.4 | 66.8 |
| CORAL (29) | 65.3±0.4 | 54.4±0.5 | 76.5±0.1 | 78.4±0.5 | 68.7 |
| MMD (30) | 60.4±0.2 | 53.3±0.3 | 74.3±0.1 | 77.4±0.6 | 66.3 |
| DANN (31) | 59.9±1.3 | 53.0±0.3 | 73.6±0.7 | 76.9±0.5 | 65.9 |
| CDANN (32) | 61.5±1.4 | 50.4±2.4 | 74.4±0.9 | 76.6±0.8 | 65.8 |
| MTL (1; 27) | 61.5±0.7 | 52.4±0.6 | 74.9±0.4 | 76.8±0.4 | 66.4 |
| SagNets (37) | 63.4±0.2 | **54.8±0.4** | 75.8±0.4 | 78.3±0.3 | 68.1 |
| ARM (28) | 58.9±0.8 | 51.0±0.5 | 74.1±0.1 | 75.2±0.3 | 64.8 |
| VREx (33) | 60.7±0.9 | 53.0±0.9 | 75.3±0.1 | 76.6±0.5 | 66.4 |
| RSC (38) | 60.7±1.4 | 51.4±0.3 | 74.8±1.1 | 75.1±1.3 | 65.5 |
| mDSDI (Ours) | **68.1±0.3** | 52.1±0.4 | 76.0±0.2 | **80.4±0.2** | **69.2** |

**Office-Home.** We have the same observation on the Office-Home dataset results in Table 8. Our mDSDI model outperforms other methods, achieving 69.2% on average. Particularly, in the Art and Real-world domain, which contains more information in the background than other domains such as a bed in the room or bike parked on the street, our model reached 68.1% and 80.4% correspondingly, significantly higher than other methods. This means that our domain-invariant features not only support generalization better but also our domain-specific ones cover helpful information in special scenarios such as backgrounds and colors related to objects in the classification task.

**Table 9:** Classification accuracy (%) on Terra Incognita.

| Method | L100 | L38 | L43 | L46 | Average |
|---|---|---|---|---|---|
| ERM (25) | 49.8±4.4 | 42.1±1.4 | 56.9±1.8 | 35.7±3.9 | 46.1 |
| IRM (18) | 54.6±1.3 | 39.8±1.9 | 56.2±1.8 | 39.6±0.8 | 47.6 |
| GroupDRO (26) | 41.2±0.7 | 38.6±2.1 | 56.7±0.9 | 36.4±2.1 | 43.2 |
| Mixup (34; 35; 36) | **59.6±2.0** | 42.2±1.4 | 55.9±0.8 | 33.9±1.4 | 47.9 |
| MLDG (11) | 54.2±3.0 | **44.3±1.1** | 55.6±0.3 | 36.9±2.2 | 47.7 |
| CORAL (29) | 51.6±2.4 | 42.2±1.0 | 57.0±1.0 | 39.8±2.9 | 47.6 |
| MMD (30) | 41.9±3.0 | 34.8±1.0 | 57.0±1.9 | 35.2±1.8 | 42.2 |
| DANN (31) | 51.1±3.5 | 40.6±0.6 | 57.4±0.5 | 37.7±1.8 | 46.7 |
| CDANN (32) | 47.0±1.9 | 41.3±4.8 | 54.9±1.7 | 39.8±2.3 | 45.8 |
| MTL (1; 27) | 49.3±1.2 | 39.6±6.3 | 55.6±1.1 | 37.8±0.8 | 45.6 |
| SagNets (37) | 53.0±2.9 | 43.0±2.5 | **57.9±0.6** | 40.4±1.3 | **48.6** |
| ARM (28) | 49.3±0.7 | 38.3±2.4 | 55.8±0.8 | 38.7±1.3 | 45.5 |
| VREx (33) | 48.2±4.3 | 41.7±1.3 | 56.8±0.8 | 38.7±3.1 | 46.4 |
| RSC (38) | 50.2±2.2 | 39.2±1.4 | 56.3±1.4 | **40.8±0.6** | 46.6 |
| mDSDI (Ours) | 53.2±3.0 | 43.3±1.0 | 56.7±0.5 | 39.2±1.3 | 48.1 |

**Terra Incognita.** Table 9 compares the classification accuracy of our mDSDI with other baselines on Terra Incognita dataset. It shows that our method provides the second-best result on the average accuracy with 48.1%, only lost to SagNet which achieves 48.6&. It is worth noticing that this dataset lacks domain-specific features to fully exploit the advantages of our algorithm. It contains

photographs of wild animals taken by camera traps; camera trap locations are different across domains. Since these cameras are static, different animals still have the same background, or in other words, there is no correlation between background and animal label in this dataset. Hence, it may lack domain-specific features, making our model rely mainly on domain-invariant features of the object animal.

**Table 10:** Classification accuracy (%) on DomainNet.

| Method | C | I | P | Q | R | S | Average |
|---|---|---|---|---|---|---|---|
| ERM (25) | 58.1±0.3 | 18.8±0.3 | 46.7±0.3 | 12.2±0.4 | 59.6±0.1 | 49.8±0.4 | 40.9 |
| IRM (18) | 48.5±2.8 | 15.0±1.5 | 38.3±4.3 | 10.9±0.5 | 48.2±5.2 | 42.3±3.1 | 33.9 |
| GroupDRO (26) | 47.2±0.5 | 17.5±0.4 | 33.8±0.5 | 9.3±0.3 | 51.6±0.4 | 40.1±0.6 | 33.3 |
| Mixup (34; 35; 36) | 55.7±0.3 | 18.5±0.5 | 44.3±0.5 | 12.5±0.4 | 55.8±0.3 | 48.2±0.5 | 39.2 |
| MLDG (11) | 59.1±0.2 | 19.1±0.3 | 45.8±0.7 | **13.4±0.3** | 59.6±0.2 | 50.2±0.4 | 41.2 |
| CORAL (29) | 59.2±0.1 | **19.7±0.2** | 46.6±0.3 | 13.4±0.4 | 59.8±0.2 | 50.1±0.6 | 41.5 |
| MMD (30) | 32.1±13.3 | 11.0±4.6 | 26.8±11.3 | 8.7±2.1 | 32.7±13.8 | 28.9±11.9 | 23.4 |
| DANN (31) | 53.1±0.2 | 18.3±0.1 | 44.2±0.7 | 11.8±0.1 | 55.5±0.4 | 46.8±0.6 | 38.3 |
| CDANN (32) | 54.6±0.4 | 17.3±0.1 | 43.7±0.9 | 12.1±0.7 | 56.2±0.4 | 45.9±0.5 | 38.3 |
| MTL (1; 27) | 57.9±0.5 | 18.5±0.4 | 46.0±0.1 | 12.5±0.1 | 59.5±0.3 | 49.2±0.1 | 40.6 |
| SagNets (37) | 57.7±0.3 | 19.0±0.2 | 45.3±0.3 | 12.7±0.5 | 58.1±0.5 | 48.8±0.2 | 40.3 |
| ARM (28) | 49.7±0.3 | 16.3±0.5 | 40.9±1.1 | 9.4±0.1 | 53.4±0.4 | 43.5±0.4 | 35.5 |
| VREx (33) | 47.3±3.5 | 16.0±1.5 | 35.8±4.6 | 10.9±0.3 | 49.6±4.9 | 42.0±3.0 | 33.6 |
| RSC (38) | 55.0±1.2 | 18.3±0.5 | 44.4±0.6 | 12.2±0.2 | 55.7±0.7 | 47.8±0.9 | 38.9 |
| mDSDI (Ours) | **62.1±0.3** | 19.1±0.4 | **49.4±0.4** | 12.8±0.7 | **62.9±0.3** | **50.4±0.4** | **42.8** |

**DomainNet.** Finally, we experiment on DomainNet, a known large-scale dataset. In Table 10, we observe that our mDSDI not only has the highest average number with $42.8\%$ but also dominates other methods in 4 domain tests. This implies that when the number of dataset increases, our method can extract more relevant information for complex tasks, such as classifying 345 classes in DomainNet. These results also mean that our model has a balance between informative domain-invariant and domain-specific features to adapt better to different environments than others, therefore showing the highest average in all settings. For instance, the "infograph" is a domain containing hundreds of words in the background (domain-specific relevant) to describe the object, even though mDSDI can not outperform the evaluation, it is still better than the many different methods. More importantly, in a sketch domain, which does not have a background link to the object, our model always achieves significantly better performance than other methods, and so implies that our domain-invariant is much better.

## C.5 Changing of Loss Functions

This appendix shows the behavior of mDSDI's losses functions during the training time. Figure 6 visualizes the changing of objective functions and their classification accuracy during training time.

We observe that the classification accuracy of domain-invariant with respect to $L_{Z_I}$ loss decreases after a few interactions, then remains stable around $30\%$ during the training time. This means that the domain-invariant extractor $Q$ fools the domain-invariant discriminator $D_I$ successfully. Regarding domain-specific features, domain-specific extractor $R$ have extracted specific features by reducing $L_{Z_S}$ loss from domain-specific classifier $D_S$. Meanwhile, the disentangle loss $L_D$ decreases to zero, showing that the features extracted from $Q$ and $R$ are disentangled. Finally, the classifier loss $L_T$ is minimized close to zero in training samples, meaning that it preserves sufficient representation for domain-invariant and domain-specific in the classification task.

## C.6 Ablation study: Important of mDSDI on the benchmark dataset

For further analysis in ablation study with PACS (2), a standard benchmark dataset in DG. This appendix studies the performance efficiency of domain-specific and domain-invariant features per each target domain under different settings mentioned in the main paper.

Table 11 shows that if the target domain is either photo, art, or cartoon, which has colors, the settings related to domain-specific features provide better performance than domain-invariant. For instance, meta-training on domain-specific (DS-Meta) reaches $87.1\%$, $79.2\%$ for the art and cartoon domain, respectively. In contrast, if the target domain is the sketch, which is without the color information, the settings related to domain-invariant outperform domain-specific features, such as $75.1\%$ for

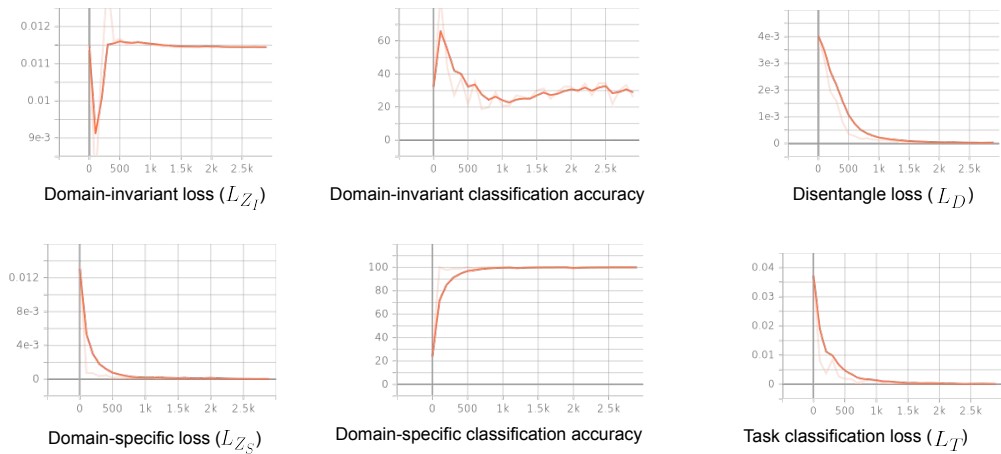

**Figure 6:** Losses visualization with tensorboard. Settings: PACS with ResNet-50, source domains include art, cartoon, sketch while target domain is photo.

**Table 11:** Classification accuracy (%) on the benchmark dataset PACS. Ablation study shows impact of domain-invariant when combined with meta-training on domain-specific in our method.

| Method | A | C | P | S | Average |
|--------|---|---|---|---|---------|
| DI | 84.5±0.6 | 77.4±0.8 | 98.0±0.3 | 74.7±1.2 | 83.7 |
| DI-Meta | 84.4±0.5 | 77.3±0.7 | 97.5±0.4 | 75.1±1.4 | 83.6 |
| DS | 85.2±0.5 | 77.9±0.9 | 98.0±0.3 | 72.5±1.3 | 83.4 |
| DS-Meta | 87.1±0.4 | 79.2±0.7 | 98.2±0.3 | 74.1±1.2 | 84.7 |
| DSDI-Without $L_D$ | 86.4±0.5 | 78.9±0.8 | 98.0±0.3 | 72.4±2.4 | 83.9 |
| DSDI-Without Meta | 84.4±0.6 | 79.2±0.8 | 98.3±0.3 | 75.7±1.4 | 84.4 |
| DSDI-Meta | 86.5±0.3 | 78.4±0.7 | **98.3±0.3** | 77.5±1.2 | 85.2 |
| DSDI-Meta DI | 84.5±0.4 | 77.8±0.8 | 97.9±0.3 | 76.6±1.1 | 84.2 |
| mDSDI-Meta DS (Ours) | **87.7±0.4** | **80.4±0.7** | 98.1±0.3 | **78.4±1.2** | **86.2** |

meta-training domain-invariant (DI-Meta). This is reasonable and confirms our assumptions that the domain-specific is color and background information while domain-invariant is the object's sketch.

Besides confirming the hypothesis of domain-specific and domain-invariant, we also observe that if we can learn both these features in the right strategy, the performance is even better. Specifically, when comparing between DSDI-Meta DI, which uses meta-training on domain-invariant, and mDSDI-Meta DS, which uses meta-training on domain-specific, mDSDI-Meta DS always achieves better performance in all target domain settings. As a result, achieved the highest on average with $86.2\%$ accuracy. It confirms the argument that, if we assume domain-invariant is stable across domains, the model should not optimize on these features. Instead, it should be used for domain-specific, which needs to be adapted on unseen domains. Meta-training on domain-invariant might lead to the model forgetting about the common because that procedure attempts to discover other independent contexts.

Finally, it is also worth noticing that without the disentanglement loss $L_D$, the average classification accuracy on target domains only achieves $83.9\%$, lower than around $2.5\%$ when compared with the highest one mDSDI-Meta DS which minimizing $L_D$. This shows that the disentangle method based on minimizing the covariance matrix in our framework work well and is essential to make domain-invariant and domain-specific independence, leading to an effective end-to-end framework to boost the generalization ability.