# OpenReview forum: "Exploiting Domain-Specific Features to Enhance Domain Generalization"
_NeurIPS.cc/2021/Conference — NeurIPS 2021 Poster_

### Official Review · Reviewer_sfZU · 2021-07-14

**Rating:** 6
**Confidence:** 5

**Summary:**

This paper proposed a meta-learning based domain generalization method that aims to extract both domain-invariant and domain-specific information of the inputs. The key idea is to apply two domain-adversarial training losses and then use cross-covariance to minimize the correlation of these two features. Experiments on several datasets have shown its effectiveness.

**Limitations And Societal Impact:**

No negative societal impact

**Main Review:**

1. Inconsistency between main idea and algorithm: The main theorem is based on mutual information, while the algorithm is based on domain-adversarial learning. However, it remains unclear why adversarial training can replace mutual information, or it is not feasible to match such algorithm with the idea. This makes it inconsistent.

2. Algorithm is not novel: it is common to disentangle the domain-invariant and domain-specific features for multi-domain learning. This algorithm is an incremental version of MLDG where they further apply two adversarial losses.

3. Missing several related work: As I stated above, there are tons of works that uses disentanglement to learning two kinds of features. However, authors did not cite or discuss with them in the paper. See the reference section.

4. Improvements are not significantly: The results of CORAL (second-best) in table 1 are only 0.6% worse than this method, but it remains much easier than this one. You should also compare their actual running time.

References:

[1] Learning Disentangled Semantic Representation for Domain Adaptation, IJCAI-19

[2] Domain Agnostic Learning with Disentangled Representations， ICML-19

[3] A Unified Feature Disentangler for Multi-Domain Image Translation and Manipulation, NeurIPS-18

[4] DISENTANGLED MULTIDIMENSIONAL METRIC LEARNING FOR MUSIC SIMILARITY, ICASSP-20

[5] Learning Causal Semantic Representation for Out-of-Distribution Prediction. arXiv 2011.01681

[6] DIVA: Domain Invariant Variational Autoencoders. PMLR 2021.

[7] Towards Better Understanding of Disentangled Representations via Mutual Information. arXiv 1911.10922

[8] A Meta-Transfer Objective for Learning to Disentangle Causal Mechanisms. ICLR-20

**Time Spent Reviewing:**

5

---

> ### Author Response · Authors · 2021-08-09
> **Official Response to Reviewer sfZU**
>
> Thanks for your feedback. We address the main concerns below.
> ### 1. Algorithms are not novel, there are tons of works that use disentanglement to learn two kinds of features. Missing several related work.
> We respectfully disagree with this comment, which might contain some misunderstanding about the main idea and technical contributions of our paper. We would like to emphasize that the goal of our paper is to introduce the use of domain-specific features together with the existing domain generalization methods that heavily rely on learning domain-invariant. To the best of our knowledge, our paper is the first work that efficiently combines domain-invariant and domain-specific in order to boost the generalization performance on the unseen target domain. We also provide a theoretical guarantee (based on the disentanglement or independence between those types of features) and convincing empirical results for our claim.
> Regarding the additional mentioned related work, please note that none of them are based on the disentanglement between domain-specific and domain-invariant features in the domain generalization setting. In particular, the methods in [1], [2], and [6] are actually based on the class and domain label disentanglement, not domain-invariant and domain-specific disentanglement. In our proposed method, the domain-specific is strongly correlated with the label (see Assumption 1) while [1], [2], and [6] make class and label independent. Meanwhile, in [5] and [8], they disentangle the causal variables, again not domain-invariant and domain-specific disentanglement. Regarding problem setting, [3] considers an unsupervised domain adaptation, and it is completely different from domain generalization. Note that in a domain generalization setting, it is strictly not allowed to access the target domain during the training process. Moreover, [4] and [7] are not even related to domain generalization settings.
>
> In Domain Generalization, utilizing domain-specific features also has been studied in the work of [9] and [10]. However, [9] introduces multiple domain-specific networks for each domain, then uses structured low-rank constraints to align them with domain-invariant. Although this encourages the better transfer of knowledge, one of its main problems is the requirement of too many domain-specific networks. Most closely related to our work is the domain-specific Masks for Generalization (DMG) framework [10], which integrates both domain-invariant and specific information; however one of its key drawbacks is that domain-invariant/domain-specific representations might not be disentangled since the learning and inferring procedures are performed implicitly (i.e., without any theoretical guarantee) through a mask generalization process. This causes both the domain-invariant features, assumed to be stable across training domains and domain-specific features, to be ineffective on the unseen domain.
>
> To handle these shortcomings, in this paper, we propose a novel Domain Generalization approach that can extract useful domain-specific information and then explicitly disentangles the domain-invariant and domain-specific representations in an efficient way without the need to train multiple networks for domain-specific. We develop a rigorous framework to formulate the elements of domain-invariant/domain-specific representations and provide necessary theoretical insights for them. Our key insight is to introduce an effective meta-optimization training framework to learn domain-specific representations from multiple training domains. Without access to any data from unseen target domains, the meta-training procedure provides a suitable mechanism to self-learn domain-specific representation.
>
> [1] Learning Disentangled Semantic Representation for Domain Adaptation, IJCAI-19
>
> [2] Domain Agnostic Learning with Disentangled Representations， ICML-19
>
> [3] A Unified Feature Disentangler for Multi-Domain Image Translation and Manipulation, NeurIPS-18
>
> [4] DISENTANGLED MULTIDIMENSIONAL METRIC LEARNING FOR MUSIC SIMILARITY, ICASSP-20
>
> [5] Learning Causal Semantic Representation for Out-of-Distribution Prediction. arXiv 2011.01681
>
> [6] DIVA: Domain Invariant Variational Autoencoders. PMLR 2021
>
> [7] Towards Better Understanding of Disentangled Representations via Mutual Information. arXiv 1911.10922
>
> [8] A Meta-Transfer Objective for Learning to Disentangle Causal Mechanisms. ICLR-20
>
> [9] Ding, Z. and Fu, Y. Deep domain generalization with structured low-rank constraint. IEEE Transactions on Image Processing, 27(1):304–313, 2018.
>
> [10] Chattopadhyay, P., Balaji, Y., and Hoffman, J. Learning to balance specificity and invariance for in and out of domain generalization, ECCV-2020.
>
> ### 2. Inconsistency between main theorem and algorithm.
> Despite not exactly minimizing measures of mutual information as discussed in information theory (which causes a high computational cost in Domain Generalization), our implementation and the theory section still have a strong connection, following these reasons:
> - The classifier within standard cross-entropy minimization in Eq. (2) and  Eq. (5) have been shown to be similar to minimal and sufficient representation with the label (definition (3)) in [11], this also a fundamental definition of the information bottleneck method which is proved in [12].
> - The domain classifiers within an adversarial training framework in Eq. (1) can be used as a proxy for the minimal and sufficient representations with domain-invariant in definition (4). Because following definition (4) in our paper, it maximizes mutual information across source domains and minimizes redundancy specific information in a particular domain. Its optimal solution is similar to the adversarial training framework in Eq. (1).
> - In the disentangle loss in Eq. (3), minimize Covariance between two random variable equivalents to minimize mutual information between them. Because, we can derive min $I(Z_I, Z_S) = min KL(P(Z_I, Z_S), P(Z_I).P(Z_S))$. Two random variables X, Y are disentangled (independent) if covariance (X, Y) = 0, hence, if they are feature vectors $Z_I = [Z_{I1}, Z_{I2}, ..., Z_{Im}]$ and $Z_S = [Z_{S1}, Z_{S2}, ..., Z_{Sn}]$, they will be disentangled (independence) if $COV(Z_{Ii}, Z_{Sj}) = 0$ for every (i, j) in the covariance matrix.
>
> [11] Yao-Hung Hubert Tsai, Yue Wu, Ruslan Salakhutdinov, and Louis-Philippe Morency. Self- supervised learning from a multi-view perspective. In International Conference on Learning  Representations, 2021.
>
> [12] Naftali Tishby, Fernando C Pereira, and William Bialek. The information bottleneck method. arXiv preprint physics/0004057, 2000.
>
> ### 3. Improvements are not significant: The results of CORAL [13] (second-best) in table 1 are only 0.6% worse than this method, but it remains much easier than this one. Should also compare their actual running time.
> It is worth noting that 0.6% is the average of 5 datasets and per each dataset, we also report the average of all domains in Table 1. It can be seen in detail the results in the Appendix B.4 that our model outperforms CORAL, especially in some domains that contain domain-specific information (background, color) such as VLCS/LabelMe (mDSDI: 68.2%, CORAL: 66.1%),Office-Home/Art (mDSDI: 68.1%, CORAL: 65.3%), Office-Home/Real-World (mDSDI: 80.4%, CORAL: 78.4%), DomainNet/Clipart (mDSDI: 62.1%, CORAL: 59.2%), DomainNet/Painting (mDSDI: 49.4%, CORAL: 46.6%), DomainNet/Photo (mDSDI: 62.9%, CORAL: 59.8%). All of these are show that our mDSDI provides better results with large margins compared to other related works.
>
> Moreover, we respectfully disagree that other methods (e.g., CORAL [13]) are much easier than our proposed method. We would like to emphasize that our model’s inference time is the almost same while its training time is much faster than CORAL [13] and many other mentioned methods during our experiments:
> - In inference time, two extractors could extract parallel, so the inference time is the same with other methods. For example, in PACS: the target domain is sketch, mDSDI and CORAL [13] have the same inference time around 15 seconds (with computing system in Appendix B.3).
> - In training time, for instance in CORAL [13], since it is based on MMD (Maximum Mean Discrepancy), it need to minimize the MMD between each pair-wise distribution in the source domains, and the computational complexity is in quadratic time w.r.t. the number of domains while the one for our model is linear. For example, in Rotated-MNIST, using the same set up, the training time for mDSDI is about 8 hours while the training of CORAL [13] takes around 14 hours when training (with a computing system in Appendix B.3).
>
> [13] Baochen Sun and Kate Saenko. Deep coral: Correlation alignment for deep domain adaptation, 2016.
>
> We hope that you can reconsider the review score. Please let us know if you would like us to do anything else.

---

> > ### Comment · Reviewer_sfZU · 2021-08-11
> > **Thanks for your response**
> >
> > I appreciate the detailed response made by the authors.
> >
> > 1. Domain-invariant and domain-specific disentanglement
> >
> > Actually, I think authors pay over attention to these two terms as they are just common existence in many previous works. I suggest that you don't emphasis on words, but on the key technologies. I acknowledge the efforts to implement this idea in DG, but it seems not very challenging. I totally understand that DG can't access the target domain data, but this does not mean there's key obstacles in bringing DA's algorithms to DG. A recent survey [Wang'21] also mentioned that learning disentanglement in DG is mostly similar to DA since their actual methodology is similar.
> >
> > Regarding the difference with the existing works, I appreciate the detailed clarifications w.r.t. each work. But I still have the following concerns:
> >
> > - [1] involved two domain classifiers, which is just built for domain disentanglement to learn domain invariant features, see its figure 3.
> >
> > - [2] did do domain disentanglement, see its Eq. (4) and corresponding paragraphs.
> >
> > - [6] learned domain and class-level representations, but I think class-invariant is also domain-specific.
> >
> > - [5] and [8] did not say the words 'domain-invariant' and 'domain-specific', but they actually did such things from the view of causality.
> >
> > - [4] and [7] were not for DG, but their main idea is quite general and can be applied to DG directly.
> >
> > Regarding meta-optimization step, it is also not novel since it is just MLDG. Finally, I'm not saying that domain-invariant and domain-specific disentanglement are not novel, but you should claim novelty on key technologies, not these general terms since everyone could have a different understand to these terms. From what I see in the methodology, the key technology is not new.
> >
> > Last but not the least, there is a AAAI-21 paper with direct claim on domain-invariant and domain-specific terms: Latent Independent Excitation for Generalizable Sensor-based Cross-Person Activity Recognition, AAAI-21. You can also compare with it.
> >
> > [Wang'21] Wang J, Lan C, Liu C, et al. Generalizing to Unseen Domains: A Survey on Domain Generalization[J]. arXiv preprint arXiv:2103.03097, 2021.
> >
> > 2. Running time
> >
> > I doubt why the computation of CORAL relies on MMD. I know that domainbed's implementation is based on MMD, but it is not the optimal. CORAL does not rely on MMD, but simple covariance computation. I suggest you can try other CORAL codes to reclaim the efficiency of your work with more accurate time comparison.
> >
> > I would like to increase my score to 4. Looking forward to your further comments.

---

> > > ### Author Response · Authors · 2021-08-12
> > > **Response to further concerns of Reviewer sfZU.**
> > >
> > > We gratefully appreciate your prompt response to our rebuttal, especially for upgrading the score for our paper. We would like to address your further concerns below.
> > >
> > > ### 1. Disentanglement methodology in DG is similar to DA.
> > > We respectfully disagree with this comment. As we mentioned in our paper and rebuttal, DG setting is often more challenging than DA because of a constraint that there is no access to the data of the target domain. We agree and are also aware that some DA-based methods such as domain adversarial neural net (DANN) or maximum mean discrepancy (MMD)-based can be applied in DG. Nevertheless, it might be hard to directly employ any approaches that work well in DA to DG. In particular, since the disentanglement technique in DA [2] relies on the source and target domain alignment, this means the training process of that approach needs access to the target data. It is not clear to employ that method in the setting of DG.
> > >
> > > ### 2. The meta-learning step is not novel since it is just MLDG.
> > > Thank you for your question. To mitigate the inaccessibility to the target domain in DG, we combine meta-learning on domain-specific with disentanglement methodology. By doing so, the model could improve generalization ability on the unseen domain by learning transferable weight representations from meta-source domains to the meta-target domain.
> > >
> > > We would like to emphasize that the meta-learning in MLDG adapts for all representation features which include domain-invariant. However, since the domain-invariant is stable across domains, pushing to adapt those features might affect the stability of those domain-invariant, leading to a lower generalization performance on the target domain. Following this idea and mitigating its drawback, we apply the meta-learning technique to exploit domain-specific only, which needs to be adapted from source domains to unseen domains to help the model perform better in the unseen target domain.
> > >
> > > ### 3. Domain-invariant and domain-specific disentanglement.
> > > Thank you for your comment. As mentioned in our rebuttal, the work in [1], [2], and [6] mainly focuses on class-relevant and domain (class-irrelevant) disentanglement. In particular, this kind of framework is based on generative modeling which tries to disentangle from the perspective of the data generation process; while our method is based on Multi-component analysis with different feature extractors (please see the detailed differences between “Multi-component analysis” and “Generative modeling” in the “Feature disentanglement-based DG” section of [Wang'21]). Similarly, in [5] and [8], we believe their causality variables imply generative mechanisms since they do not provide any explicit definitions for “domain-invariant” and “domain-specific” features.
> > >
> > > Moreover, it is worth noting that, according to [Wang'21], some of these papers (e.g., [2], [5], and [6]) have been also mentioned and categorized into the group of “Generative modeling” disentanglement approaches.
> > >
> > > ### 4. Comparison with: Latent Independent Excitation for Generalizable Sensor-based Cross-Person Activity Recognition, AAAI-21.
> > > Thank you for your suggestion. We agree that this work disentangles domain-invariant and domain-specific features and can be viewed as an instance of “Multi-component analysis” disentanglement. However, it is worth noting that the main idea of this paper is to try to disentangle domain-invariant and domain-specific at training time, then discard domain-specific information at inference time. Our insight is that domain-specific should not be removed at inference time (please see in the example classifying dog or fish images from two source domains: sketch and photo (lines 27-37 in the main paper)). The theoretical guarantee for the benefit of using the domain-specific is also mentioned and verified in Theorem 1 - we also would like to emphasize that it is one of the main theoretical contributions of our paper.
> > >
> > > ### 5. Running time
> > > Thank you for your question. Our apologies for the misleading. We do not mean that CORAL computes K Gaussian kernels for the representations as MMD does. What we really want to state is that CORAL is similar to MMD because both of them rely on minimizing pairwise distances of the distributions in the source domains. It is clear that their computational complexity is quadratic w.r.t. the number of domains; while the one for our algorithm is linear.
> > >
> > > Regarding the code, it can be seen that implementations in CORAL and MMD are very similar except for the kernel type (please see lines 609-626 in mDSDI/DomainBed/domainbed/algorithms.py provided in our supplementary). As requested, we have checked the original CORAL source code, however, since this code is based on MATLAB, it is hard to be compared with our python code.
> > >
> > > Thank you once again for your constructive feedback. Please let us know if you still have any concerns.

---

> > > > ### Comment · Reviewer_sfZU · 2021-08-14
> > > > **Most of my concerns are resolved**
> > > >
> > > > As the title. Increased my score to 6. Happy weekend!

---

### Official Review · Reviewer_FWcu · 2021-07-16

**Rating:** 7
**Confidence:** 4

**Summary:**

This paper studies the importance of relying on both domain-specificity (DS) and domain-invariance (DI) in domain generalization (DG) problems. It formalizes the problem through the lens of information theory - more specifically, through the lens of the information bottleneck theory. It advocates for learning both DS and DI representations, and proposes a well motivated method that accommodates this need. Results on the standardized DomainBed benchmark are strong, and proper ablation studies are provided.

**Limitations And Societal Impact:**

No. It would be valuable a discussion about possible situations in which related works perform better. For example, can the Authors imagine some situations where instead learning just DI representations is a better choice? For what concerns societal impact, DG's goal is to make machine learning systems safer when deployed in the wild, hence something along these lines may be proper.

**Main Review:**

I enjoyed reading this paper: it proposes a well-formulated analysis on the importance of relying on both DS and DI representations in domain generalization. The proposed ideas are well exposed both from a theoretical perspective and from a more practical view. While some parts result rather mathematically heavy for "more empirical" readers, the Authors made the effort of making their insights accessible to the broader audience - for example, I believe Figure 1 is a very effective one.

The proposed method is reasonable, and supported by strong results on a recently released testbed - DomainBed. The importance of each contribution is supported by a detailed ablation study (Sec. 3.4). While the exploitation of both DS and DI representations is not novel (previous works have been properly cited), I believe that this work brings enough contributions to consider the proposed approach original.

I think the work could be improved in a couple of directions:

- After having finished Sec. 2.2, I was expecting to find a method more connected with information theory. It is true that domain classifiers within an adversarial training framework can be used as a proxy for the Mutual Information, but some more sound approaches would be applicable, too. For example, one could use MINE [Belghazi et al., ICML 2018] to both estimate the MI between two sources and directly optimizing w.r.t it - since the module is fully differentiable. Have the Authors thought about narrowing the gap between theory and implementation?

- I have found the disentanglement part very difficult to follow. What does it mean "the disentanglement condition between two random vectors $Z_I$ and $Z_S$ can be solved by forcing their covariance matrix [...]"? Is it possible to explain this intuition in other terms, since we still need to converge towards a proper definition of "disentanglement"?

Summarizing, I believe this is a good work, that can definitely be interesting for the NeurIPS community. I look forward reading the Author response and discussing with other Reviewers.

Some minor comments below:

- Should Definition 2 not also take into account the labeling? Domain specificity is an interesting property if the distributions are different within the same classes, otherwise it can be seen as a trivial property.

- How realistic is Assumption 1 in practice? I think it is worth a couple of lines.

- Line 129: "learning procedure"?

- Meta-learning could be effective for both DS and DI, why just DS?

**Time Spent Reviewing:**

/

---

> ### Author Response · Authors · 2021-08-08
> **Official Response to Reviewer FWcu**
>
> Thank you for your detailed reviews and like our VennDiagram with Domain Generalization from an Information Bottleneck perspective. We truly appreciate that. We would clarify your concerns below and make changes in the revision version as you suggest.
>
> ### 1. Narrowing the gap between theory and implementation.
> We agree with you that there is a narrow gap between our theory and implementation. However, despite not exactly minimizing measures of mutual information as discussed in information theory (which causes a high computational cost in Domain Generalization), our implementation and the theory section still have a strong connection, following these reasons:
> - The classifier within standard cross-entropy minimization in Eq. (2) and  Eq. (5) have been shown to be similar to minimal and sufficient representation with the label (definition (3)) in [1], this also a fundamental definition of information bottleneck method which is proved in [2].
> - The domain classifiers within an adversarial training framework in Eq. (1) can be used as a proxy for the minimal and sufficient representations with domain-invariant in definition (4). Because following definition (4) in our paper, it maximizes mutual information across source domains and minimizes redundancy specific information in a particular domain. Its optimal solution is similar to the adversarial training framework in Eq. (1).
> - In the disentangle loss in Eq. (3), minimize Covariance between two random variable equivalents to minimize mutual information between them. Because, we can derive min $I(Z_I, Z_S) = min KL(P(Z_I, Z_S), P(Z_I).P(Z_S))$. Two random variables X, Y are disentangled (independent) if covariance (X, Y) = 0, hence, if they are feature vectors $Z_I = [Z_{I1}, Z_{I2}, ..., Z_{Im}]$ and $Z_S = [Z_{S1}, Z_{S2}, ..., Z_{Sn}]$, they will be disentangled (independence) if $COV(Z_{Ii}, Z_{Sj}) = 0$ for every (i, j) in the covariance matrix.
>
> [1] Yao-Hung Hubert Tsai, Yue Wu, Ruslan Salakhutdinov, and Louis-Philippe Morency. Self- supervised learning from a multi-view perspective. In International Conference on Learning  Representations, 2021.
>
> [2] Naftali Tishby, Fernando C Pereira, and William Bialek. The information bottleneck method. arXiv preprint physics/0004057, 2000.
>
> ### 2. What is “"the disentanglement condition between two random vectors $Z_I$ and $Z_S$ can be solved by forcing their covariance matrix [...]"? ”Explain disentanglement intuition in other terms.
> It means that we minimize the Covariance Matrix between two random vectors $Z_I$ (domain-invariant) and $Z_S$ (domain-specific) with the zero matrix (see Eq. (3) in line 197). If so, we can explicitly decompose these features. Because mathematically, two random variables X, Y are disentangled (independent) if covariance (X, Y) = 0. If they are feature vectors $X = [X_1, X_2, ..., X_m]$ and $Y = [Y_1, Y_2, ..., Y_n]$, they will be disentangled (independence) if $COV(X_i, Y_j) = 0$ for every (i, j) in the covariance matrix.
>
> ### 3. Should Definition 2 not also take into account the labeling?
> Yes, we agree Definition 2 should take labeling into account. However, due to covariate shift assumption and in our implementation, we only apply standard supervised learning to minimize cross-entropy to optimize $Z_S$ (lines 191-193). If we follow conditional alignment (label shift assumption) as you suggested in Definition 2, it requires other complex techniques which also consume higher resource computation, so we ignore them here to concentrate on our main contributions.
>
> Indeed, instead of minimizing the covariance matrix between $Z_I$ and $Z_S$, we have also tried adversarial training to minimize the mutual information between $Z_I$ and $Z_S$, this will reduce computational cost, and then add conditional alignment techniques as you suggested. The idea is we can derive $min I(Z_I, Z_S) = min KL(P(Z_I, Z_S), P(Z_I).P(Z_S))$, and if we shuffle $Z_S$ respected to index of samples in each mini-batch, we also can disentangle these features without being affected by feature dimension. Then, we add conditional alignment by minimizing entropy regularization in [3]. We run in PACS and have the following results: “85.7 ± 0.5” for Art, “78.9 ± 0.8” for Cartoon, “97.1 ± 0.3” for Photo, “77.4 ± 2.4” for Sketch. Because this result is insignificant (might be because of an unstable adversarial training technique), we decided to follow the covariate shift assumption made, and use the Covariance matrix minimization technique.
>
> Thanks for your suggestion, in the future, we will continue to try to replace covariance matrix optimization by tuning adversarial training to disentangle these two features, and retry conditional alignment techniques as you suggested.
>
> [3] S. Zhao, M. Gong, T. Liu, H. Fu, and D. Tao. Domain generalization via entropy regularization. In Advances in Neural Information Processing Systems, volume 33, 2020.
>
> ### 4. How realistic is Assumption 1 in practice?
> It is plausible if you see our mentioned example about fish and dog classification in lines 118 to 121. Furthermore, you could also see several examples in our experiments. For instance, in the benchmark dataset DomainNet (lines 267-268 in the main paper), in real-world domain, many bed pictures contain a bed in the room or bike pictures have bicycles parked on the street, our model also achieve significant higher results than other methods (see Tab.7 in Appendix B.4). Other examples are in PACS such as dogs in the yard or guitars lying on a table in photo and art domains. These examples are strongly related to assumption 1 (Label-correlated domain-specificity) in which specific information correlates with labels for a particular domain.
>
> ### 5. Meta-learning could be effective for both domain-specific and domain-invariant, why just domain-specific?
> As claimed in footnote 1. We do not think that meta-learning is effective for domain-invariant because the intuition of meta-learning is using gradient update to learn transferable weight representations from meta-source domains to quickly adapt to the meta-target domain. However, domain-invariant is stable across domains, pushing to adapt might make the model ignore their information to concentrate on the independent context of each domain.
>
> Therefore, our key insight is to introduce an effective meta-optimization training framework to learn domain-specific representations from multiple training domains. Without access to any data from unseen target domains, the meta-training procedure provides a suitable mechanism to self-learn domain-specific representation. As a result, we only apply for the domain-specific to support transferable useful domain-specific representations.

---

> > ### Comment · Reviewer_FWcu · 2021-08-12
> > **Confirm original rating**
> >
> > Thank you for your detailed response. I confirm my initial rating on this paper.
> >
> > I would like to encourage the Authors to expand on the discussion about theory/implementation gap in the final version of their paper, since the same concern has been raised in different reviews.

---

> > > ### Author Response · Authors · 2021-08-12
> > > **Thank you**
> > >
> > > Dear Reviewer,
> > >
> > > Thank you very much once again for your great feedback and encouragement.
> > > We will definitely take it into account and will add a more detailed discussion to clarify that theory/implementation gap in the revised version of our paper.
> > >
> > > All the best,
> > >
> > > Authors.

---

### Official Review · Reviewer_xiMp · 2021-07-17

**Rating:** 7
**Confidence:** 4

**Summary:**

The article describes an approach for domain generalization (DG) that exploits domain-specific features. In particular, the model uses two feature extractors, one extracting domain-aligned features (adversarially learned), and one extracting domain-specific ones (by minimizing a domain classification loss). To enforce that the extracted domain-specific features are useful for unseen target domains, a meta-learning procedure is adopted. Experiments on the recent DomainBed benchmark show that the approach outperforms the state of the art on average, achieving consistently top or comparable results across all datasets.

**Limitations And Societal Impact:**

The article does not discuss either limitation of the approach or possible societal impact. For the latter, there are not clear impacts (other than performance guarantees in non-tested scenarios), but limitations of the model (e.g. need of domain labels) should be better discussed in the conclusion.

**Main Review:**

I find the method described in the paper interesting, exploiting both the information shared across domains (through the domain-aligned features) and generalizable domain-specific information (through the domain-specific features extractor). The results clearly show the effectiveness of the approach in the challenging DomainBed benchmark. The article has some weaknesses related to referencing relation to previous works and comparisons with closer baselines. However, I do not deem them to be major, thus I lean toward a positive score. Below I detail my comments.

Strengths:
+ While the use of domain-aligned features has been widely explored in the DG literature (e.g. [2,3]) it is still unclear how to make effective use of domain-specific ones. Previous works integrated them through ensembles (e.g. [43]) or meta-learning techniques (e.g. [11,a]). However, few papers consider using both domain-aligned and domain-specific features (e.g. [7]). This paper is an effective step in this direction, learning to integrate domain-specific features into domain-aligned ones through a meta-learning procedure.

+ The results on DomainBed are impressive in some challenging scenarios. For instance, for DomainNet the proposed approach achieves 42.8% accuracy vs 41.5 of the best competitor. Also in Office-Home, the advantage over the best competitor is +0.5% and in VLCS (here ERM already is a strong baseline, due to the limited domain-shift), the method improves the best competitor by more than 1%.

+ Analyses (e.g. Fig. 3, Table 2) clearly show the impact of each of the proposed components.

Weaknesses:

1. The paper is closely related to [7], which also tries to exploit both domain-specific and domain-aligned features. However, [7] is not present in the experimental comparisons, similarly to [43], another effective approach that uses an ensemble of domain-specific parameters. While they are not present in DomainBed, performing comparisons in their same settings and splits (e.g. PACS [7,43] following [2], Office-Home [43], DomainNet [7]) would strengthen the results, showing that the proposed approach is actually the state of the art for exploiting domain-specific features.

2. Following on the previous, the description and relations with the most related works in the DG literature is limited to the final part of the introduction. It would have been helpful to add a specific section to highlight both the existing domain alignment methods (e.g. [2,3]) and domain-specific architectures [43] as well as methods exploiting both specific and agnostic features [7] and meta-learning techniques [11]. Clearly describing the related works and the differences with the proposed approach is fundamental to highlight the contributions and should not be limited to few paragraphs of the intro (given that a reader may even not be familiar with the DG problem).

3. In Table 3, the ablation study is conducted in a dataset created by the authors. This choice may seem suspicious to readers since the same dataset is not present in the experimental analysis and may appear specifically created for the ablation study. I would suggest either incorporate the dataset into the wider experimental comparisons of Table 1 or replacing Table 2 with the same ablation study but conducted on PACS (Table 9 of the supplementary).

4. Definitions 1 and 2, as well as the alignment objective of Eq. (1), consider only the marginal input distribution, without considering the joint distribution of input and labels. This assumes the presence of a covariate-shift among the domains which may actually be limiting in practice due to the strong assumptions needed for achieving a good cross-domain alignment [b]. Have the authors considered applying conditional alignment techniques to improve the results?

5. Footnote 1, describing the differences of the approach w.r.t. MLDG [11] should be integrated into the main text, to better highlight the contribution of the approach w.r.t. existing meta-learning techniques.

References:

[a] Li, Da, et al. "Episodic training for domain generalization." ICCV 2019.

[b] David, Shai Ben, et al. "Impossibility theorems for domain adaptation." International Conference on Artificial Intelligence and Statistics 2010.

**Time Spent Reviewing:**

4

---

> ### Author Response · Authors · 2021-08-08
> **Official Response to Reviewer xiMp**
>
> Thank you for your detailed and constructive reviews. We truly appreciate that. We address your concerns below and make changes to the revision version of our paper.
>
> ### 1. The first section should be divided into two different sections: introduction and related works, footnote 1 should appear in the main text, and should add related works under the DomainBed setting.
> Thanks for your suggestions, we will take them into account in the revised version of our paper.
>
> ### 2. Original Colored-MNIST dataset is left out while creating a different one with the same name, incorporating the dataset into the wider experimental comparisons of Table 1.
> Thank you very much for pointing that out. We apologize for the misleading name. We will then rename them to “Background-Color-MNIST” and follow your suggestion to incorporate the original dataset into the wider experimental comparisons of Table 1.
>
> As requested, we provide our additional comparison after quickly re-implementing ERM and IRM on our Colored-MNIST dataset, and the results are provided below:
>
> |Method|Accuracy|
> |-|-|
> |ERM|76.2±3.6|
> |IRM|66.7±3.2|
> |DI|65.7±4.6|
> |DI-Meta|63.6±5.1|
> |DS|70.7±4.8|
> |DS-Meta|75.3±3.4|
> |DSDI-Without Meta|80.4±1.7|
> |DSDI-Meta|82.1±1.4|
> |DSDI-Meta DI|79.0±2.3|
> |mDSDI-Meta DS (Ours)|**89.7±0.8**|
>
> These results are much lower than our mDSDI and quite similar with our baselines including "DI", "DI-meta", "DS", and "DS-meta". The reason is that, for both ERM and IRM, they do not take both domain-invariant and domain-specific into account, they only use one Resnet-50 feature extractor. Meanwhile, our mDSDI integrates both two different feature extractors to handle domain-invariant and domain-specific, producing a high performance boost. Besides the model complexity reason, other components in our loss (see Sec. (2.3) and Sec. (3.4)) also boost its performance significantly. For instance, DSDI-Without Meta also uses two feature extractors, however without meta-training on domain-specific, its results are lower than mDSDI by around 9% in Table 2.
>
> ### 3. Considered applying conditional alignment techniques to improve the results.
> Thank you for your suggestion. Yes, we have considered this to relax our assumption (on the covariate shift) at submission time. However, since the current conditional alignment techniques (e.g., [1] and [2]) apparently require additional computation cost; this can increase the computational complexity of our algorithm. We will definitely try to find an appropriate way to incorporate two of them in a unified framework in the future work of our paper.
>
> Instead of minimizing the covariance matrix between $Z_I$ and $Z_S$, we have also tried the adversarial training to minimize the mutual information between $Z_I$ and $Z_S$ (to reduce the computational cost) and then combine with conditional alignment techniques. The idea is that we can derive $\min I(Z_I, Z_S) = \min KL(P(Z_I, Z_S), P(Z_I). P(Z_S))$, and if we shuffle $Z_S$ w.r.t. to the index of samples in each mini-batch, we also can disentangle these features without being affected by feature dimension. Then, we add conditional alignment by minimizing entropy regularization in [2]. We run in PACS and have the following results: “85.7 ± 0.5” for Art, “78.9 ± 0.8” for Cartoon, “97.1 ± 0.3” for Photo, “77.4 ± 2.4” for Sketch. Because those results are insignificant (maybe because of an unstable adversarial training technique), we decided to follow the covariate shift assumption made, and use the Covariance matrix minimization technique.
>
> We also plan to replace covariance matrix optimization by tuning adversarial training to disentangle these two features and also employ conditional alignment techniques.
>
> [1] David, Shai Ben, et al. "Impossibility theorems for domain adaptation." International Conference on Artificial Intelligence and Statistics 2010.
>
> [2] S. Zhao, M. Gong, T. Liu, H. Fu, and D. Tao. Domain generalization via entropy regularization. In Advances in Neural Information Processing Systems, volume 33, 2020.

---

> > ### Comment · Reviewer_xiMp · 2021-08-20
> > **Thanks for the response**
> >
> > I thank the authors for the extensive response provided to address the concerns I and the other reviews raised. I find the response to address my concerns and the major ones raised by the other reviewers, thus I am increasing my rating to 7.

---

### Official Review · Reviewer_SAJX · 2021-07-25

**Rating:** 4
**Confidence:** 5

**Summary:**

This work claims that the domain-specific knowledge also can make contributions to the classification of unseen target images in domain generalization setting, and attempts to prove the point from information theory. Moreover, this work develops the corresponding algorithm to solve domain generalization and carries out experiments on several benchmarks by comparing with many baselines.

**Limitations And Societal Impact:**

The major concern is the limitation of application based on the proposed idea.

**Main Review:**

This work claims that the domain-specific knowledge also can make contributions to the classification of unseen target images in domain generalization setting, and attempts to prove the point from information theory. Moreover, this work develops the corresponding algorithm to solve domain generalization and carries out experiments on several benchmarks by comparing with many baselines.

However, the major concern is the limitation of application based on the proposed idea.  Concretely, can this idea also work for the situation where the domain-specific information (background, color) of unseen target images is totally different from that of multiple source domains? In addition, although the theoretical analyses illustrate the domain-specific knowledge involves semantic information related to category, these analyses do not explain why the application of domain-specific knowledge can improve the generalization ability of model. Moreover, it is not clear why the developed algorithm can explicitly decompose the features into domain-specific representation and domain-invariant ones. Finally, it would be better to compare this work with the recent work [1] on domain generalization to verify the  advantage of the proposed method.

[1] Xu, Qinwei, et al. "A Fourier-based Framework for Domain Generalization." Proceedings of the IEEE/CVF Conference on Computer Vision and Pattern Recognition. 2021.

**Time Spent Reviewing:**

6 hours

---

> ### Author Response · Authors · 2021-08-08
> **Official Response to Reviewer SAJX**
>
> Thanks for your feedback. We address the main concerns below.
> ### 1. Can this idea also work for the situation where the domain-specific information (background, color) of unseen target images is totally different from that of multiple source domains?
> If domain-specific features of the target domain never appear in source domains, it will be very hard (even impossible) for existing methods to extract those features. Our method focuses on extracting the domain-specific features that appear in the source domains. In most practical situations, these features, if exist in the target domain, are consistent between source and target domains. That assumption is shown to be a necessary condition for a DA/DG model to learn useful features for the generalization in the target domain [2]. The motivation of our proposed method is that the “Domain Specific (DS) information could aid the generalization performance, especially when the number of source domains increases (lines 26)”, and in assumption 1 that “a domain-specific representation $Z_{S}^{(1)}$ can be extracted by the deterministic mapping from source domain $X^1$: $Z_{S}^{(1)} = R(X^1)$”.
>
> [2] David, S.B., Lu, T., Luu, T. and Pál, D., 2010, March. Impossibility theorems for domain adaptation. In Proceedings of the Thirteenth International Conference on Artificial Intelligence and Statistics (pp. 129-136). JMLR Workshop and Conference Proceedings.
>
> ### 2. Theoretical analyses do not explain why the application of domain-specific knowledge can improve the generalization ability of a model.
> Thanks for your question. The theoretical guarantee for the benefit of using the domain-specific is mentioned and verified in Theorem 1 - we also would like to emphasize that it is one of the main theoretical contributions of our paper. In particular, Theorem 1 indicates that the sum of mutual information between domain-invariant $I(Z_I^*, Y)$ and domain-specific value $\epsilon_1$ is always larger than the one for domain-invariant $I(Z_I^*, Y)$ only (please see the details of our proof in Appendix A.2). Consequently, based on the definition of the Information Bottleneck method, the higher the mutual information with label Y, the better the prediction (higher generalization ability) is [3].
>
> [3] Naftali Tishby, Fernando C Pereira, and William Bialek. The information bottleneck method. arXiv preprint physics/0004057, 2000.
>
> ### 3. Not clear why the algorithm can explicitly decompose the features into domain-specific representation and domain-invariant ones.
> Thanks for asking for clarification on that. The separation (i.e., disentanglement or independence) between domain-invariant and domain-specific features is one of the key ideas of our proposed method. We’ve already explained how to decompose these features in Sec. (2.3) (lines 194-197, and Eq. (3)) of the main paper. More specifically, these features can be decomposed explicitly by minimizing the Covariance Matrix between two random vectors $Z_I$ (domain-invariant) and $Z_S$ (domain-specific). Mathematically, two random variables X and Y are disentangled (independent) if the covariance between X and Y equals 0. In case they are feature vectors $X = [X_1, X_2, ..., X_m]$ and $Y = [Y_1, Y_2, ..., Y_n]$, they will be disentangled (independence) when $COV(X_i, Y_j) = 0$ for every (i, j)-component in the covariance matrix.
>
> ### 4. Adding "A Fourier-based Framework for Domain Generalization [1]" for experimental comparison.
> Thank you for your suggestion. As requested, we provide our additional comparison after re-implementing the work of “A Fourier-based Framework for Domain Generalization [1]” on PACS dataset, and the results are provided below:
>
> |Method|Art|Cartoon|Photo|Sketch|Average|
> |-|-|-|-|-|-|
> |ERM|84.7±0.4|**80.8±0.6**|97.2±0.3|79.3±1.0|85.5|
> |IRM|84.8±1.3|76.4±1.1|96.7±0.6|76.1±1.0|83.5|
> |GroupDRO|83.5±0.9|79.1±0.6|96.7±0.3|78.3±2.0|84.4|
> |Mixup|86.1±0.5|78.9±0.8|97.6±0.1|75.8±1.8|84.6|
> |MLDG|85.5±1.4|80.1±1.7|97.4±0.3|76.6±1.1|84.9|
> |CORAL|88.3±0.2|80.0±0.5|97.5±0.3|78.8±1.3|86.2|
> |MMD|86.1±1.4|79.4±0.9|96.6±0.2|76.5±0.5|84.6|
> |DANN|86.4±0.8|77.4±0.8|97.3±0.4|73.5±2.3|83.6|
> |CDANN|84.6±1.8|75.5±0.9|96.8±0.3|73.5±0.6|82.6|
> |MTL|87.5±0.8|77.1±0.5|96.4±0.8|77.3±1.8|84.6|
> |SagNets|87.4±1.0|80.7±0.6|97.1±0.1|**80.0±0.4**|**86.3**|
> |ARM|86.8±0.6|76.8±0.5|97.4±0.3|79.3±1.2|85.1|
> |VREx|86.0±1.6|79.1±0.6|96.9±0.5|77.7±1.7|84.9|
> |RSC|85.4±0.8|79.7±1.8|97.6±0.3|78.2±1.2|85.2|
> |Fourier [1]|85.1±0.4|80.6±0.6|97.2±0.4|79.5±1.2|85.6|
> |mDSDI|**87.7±0.4**|80.4±0.7|**98.1±0.3**|78.4±1.2|86.2|
>
> This doesn’t exceed the corresponding results of our mDSDI and significantly lower than what the corresponding paper reported. The reason is that they follow JiGen [4] and RSC [5] experimental settings while we are following DomainBed [6]. There are a number of differences between these settings, such as data in the validation/testing split, number of iterations, number of steps to evaluation, model selection techniques, and more (see Appendix B.3 of our paper and Sec. (4.2) of [1]), all of these will affect experimental results in Domain Generalization.
>
> However, JiGen [4] and RSC [5] settings seem unfair for all other methods such as ERM. They use several tricks, e.g., data-augmentation and more training iterations, that boost their reported results. These critical issues have been raised in the Domain Generalization community recently (e.g., https://github.com/DeLightCMU/RSC/issues/12). It is the main motivation for the DomainBed [6] paper. Therefore, we only follow DomainBed [6] settings.
>
> [1] Xu, Qinwei, et al. "A Fourier-based Framework for Domain Generalization." Proceedings of the IEEE/CVF Conference on Computer Vision and Pattern Recognition. 2021.
>
> [4] Fabio M Carlucci, Antonio D’Innocente, Silvia Bucci, Barbara Caputo, and Tatiana Tommasi. Domain generalization by solving jigsaw puzzles. In Proceedings of the IEEE Conference on Computer Vision and Pattern Recognition, pages 2229–2238, 2019.
>
> [5] Zeyi Huang, Haohan Wang, E. Xing, and D. Huang. Self-challenging improves cross-domain generalization. In ECCV, 2020.
>
> [6] Gulrajani, I., & Lopez-Paz, D. (2020). In Search of Lost Domain Generalization. In the International Conference on Learning Representations.
>
> We hope that you can reconsider the review score. Please let us know if you would like us to do anything else.

---

> > ### Author Response · Authors · 2021-08-29
> > **Waiting for your response**
> >
> > Dear Reviewer,
> >
> > Thanks a lot for your efforts in reviewing this paper. We have tried our best to address all your concerns and provided clarifications on all confusing concepts.
> >
> > The discussion period is nearing an end, thus we wonder if you can spend some time going over our newest replies, to see if they successfully answered your questions or not. This is also to give us a decent amount of time to address any of your remaining/additional concerns.
> >
> > Best regards,
> >
> > Authors.

---

### Author Response · Authors · 2021-08-25
**Look forward to the final feedback before the discussion period ends**

Dear reviewers,

We would like to thank you again for spending your time reading the rebuttals as well as evaluating our paper. We truly appreciate that.

As the ending of the discussion period is approaching, we look forward to hearing your feedback if our answers haven’t addressed your questions. We would be happy to discuss if you still have any concerns.

All the best,

Authors.

---

### Decision · Program_Chairs · 2021-09-28

**Decision:**

Accept (Poster)

**Comment:**

This paper received primarily positive reviews. The authors provided sufficient additional details during the rebuttal to reduce most concerns about the work. There was originally some concerns about the gap between the presented theory and the implemented algorithm reiterated by a few reviewers. The authors should be sure to include their clarifications presented during the discussion phase in the final version of their paper. The authors should note that the AC has read the concerns brought up by SAJX as well as the author rebuttal and is convinced by the responses provided. Once again, please do provide the discussions and clarifications in the final version. There was substantial new clarifying information during the discussion phase and this is needed to be included in the final draft.

**Consistency Experiment:**

NeurIPS has a long history of experimentation. In 2014, NeurIPS ran an experiment in which 10% of submissions were reviewed by two independent committees to quantify the randomness in the review process. This year, we repeated a variant of this experiment to see how the quality of the review process has changed over time.  This paper was part of the experiment and was therefore assigned to two committees (consisting of reviewers, an Area Chair, and a Senior Area Chair) that reached independent decisions.  If both committees made the same recommendation, this recommendation was followed. If a single committee recommended acceptance, the paper was accepted (with the exception of a few cases in which the other committee identified what we considered a fatal flaw, e.g., an error in a key result).

This copy’s committee reached the following decision: **Accept (Poster)**

The other committee assigned to the paper recommended **Reject**.  You can find the other set of reviews, along with any follow up discussion with the authors here:
https://openreview.net/forum?id=vKxFYApxBjr